# Liquid-liquid reactions performed by cellular reactors

Jinzhe Cao [1] & Shengyang Tao [1,2,3,4] ✉

Liquid-liquid reactions play a significant role in organic synthesis. However, control of the phase interface between incompatible two-phase liquids remains challenging. Moreover, separating liquid acid, base and oxidants from the reactor takes a long time and high cost. To address these issues, we draw inspiration from the structure and function of cells in living organisms and develop a biomimetic 3D-printed cellular reactor. The cellular reactor houses an aqueous phase containing the catalyst or oxidant while immersed in the organic phase reactant. This setup controls the distribution of the phase interface within the organic phase and increases the interface area by 2.3 times. Notably, the cellular reactor and the aqueous phase are removed from the organic phase upon completing the reaction, eliminating additional separation steps and preventing direct contact between the reactor and acidic, alkaline, or oxidizing substances. Furthermore, the cellular reactor offers the advantages of digital design feasibility and cost-effective manufacturing.

Liquid-liquid reactions are vital in chemical synthesis due to the fast mass transfer between interfaces, uniform temperature field distribution, dynamic interfacial composition updating, and resistance to catalyst poisoning[1–5]. Typically, one reactant acts as a dispersed phase, forming tiny droplets in the continuous phase of another substance, increasing the area for molecular diffusion. The surface tension causes the dispersed-phase droplets to adopt a spherical shape, minimizing the system's energy. However, collisions between droplets lead to reduced contact between the liquid phases, often necessitating fast mechanical stirring or emulsifiers.

Stirred reactors are widely used in chemical engineering and industrial processes for liquid-liquid reactions[6–9]. Stirring facilitates thorough mixing of the two phases, enhancing reaction rates[10]. It also helps maintain temperature uniformity, preventing temperature gradients due to exothermic or endothermic reactions[11]. However, stirred reactors have drawbacks, such as high shear forces that can damage shear-sensitive chemicals[12] and insufficient stirring speeds that fail to maintain adequate interfacial area between the two liquid phases, leading to reduced reaction rates.

Using emulsifiers stabilizes droplets and reduces their size, thereby increasing the total interfacial area[13–15]. However, the compatibility of emulsifiers with the reaction system must be considered, as the introduction of inappropriate emulsifiers can decrease reaction efficiency. Additionally, the resulting emulsion complicates the separation and purification of products post-reaction. Most importantly, the use of emulsifiers inevitably requires rapid mechanical stirring.

Phase transfer catalysts can enhance the efficiency of liquid-liquid biphasic reactions without the need for vigorous stirring[16]. However, the subsequent separation process becomes more complex due to the high affinity of phase transfer catalysts for both phases[17]. Therefore, designing a novel reactor that can enhance liquid-liquid interfacial mass transfer at low stirring speeds and allow for easy separation after the reaction is crucial.

In nature, numerous liquid-liquid reactions exhibit remarkable controllability[18]. Cell membranes represent a widely occurring natural interface for such reactions, where reactants diffuse to the interface, and products depart, maintaining interface stability throughout the process[19]. Syncytia, large cellular structures formed by the fusion of multiple cells[20], facilitate intercellular cooperation and functional integration, enhancing cellular stability and efficiency. Inspired by the structure of syncytia and combining this with the structure of stirred

[1]School of Chemistry, Dalian University of Technology, 116024 Dalian, Liaoning, China. [2]State Key Laboratory of Fine Chemicals, Dalian University of Technology, 116024 Dalian, Liaoning, China. [3]Frontier Science Center for Smart Materials, Dalian University of Technology, 116024 Dalian, Liaoning, China. [4]Dalian Key Laboratory of Intelligent Chemistry, Dalian University of Technology, Dalian, Liaoning, China. ✉ e-mail: taosy@dlut.edu.cn

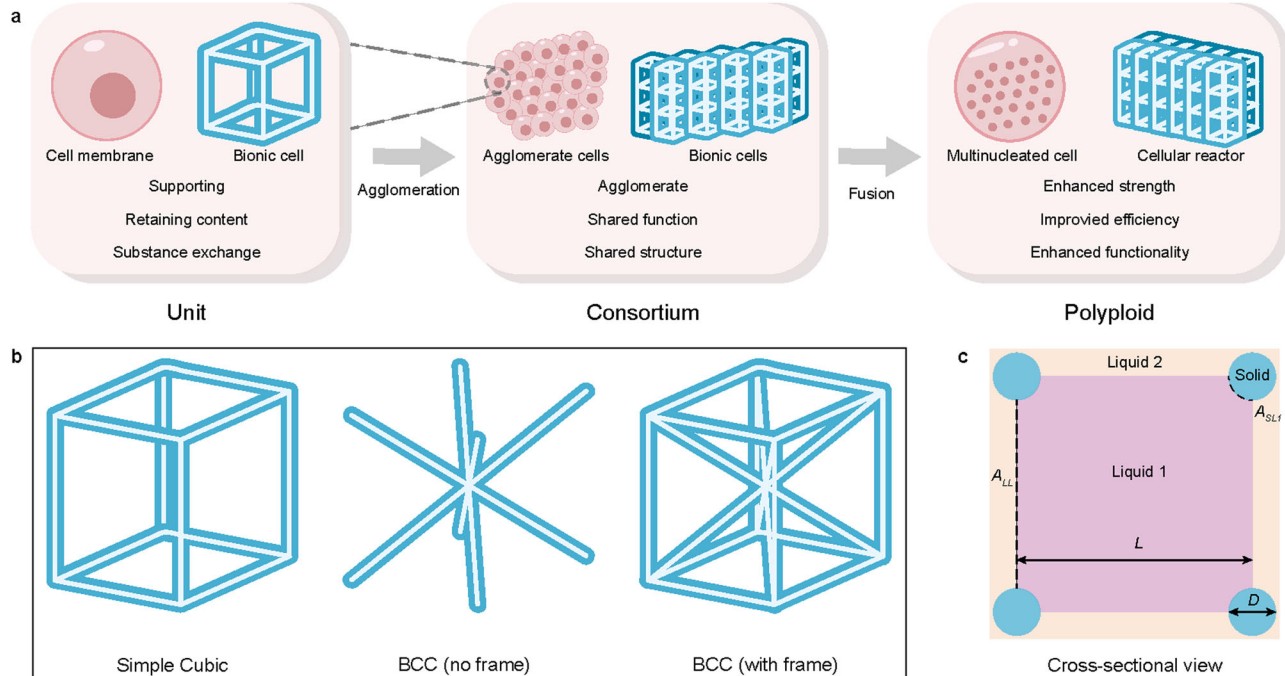

**Fig. 1 | Design of cellular reactors. a** Schematic structure of multinuclear cell and cellular reactor. **b** Three cell unit types used in this study. **c** Working cross-section of a cellular unit. Liquids 1 and 2 respectively represent the two-phase liquids located inside and outside the cellular reactor. *D* represents the diameter of the cell truss, *L* is the cell length, and $D^*$ is a characteristic dimension of the cell. Porosity, $\varepsilon$ is another dimensionless number that reflects the percentage of cells' capacity for holding liquid. $A_{SL1}^* = \frac{A_{SL1}}{L^2}$, and $A_{LL}^* = \frac{A_{LL}}{L^2}$ are dimensionless numbers relating the solid-liquid1 and the liquid-liquid contact areas, respectively.

reactors, we propose a type of cellular reactor. This design merges multiple units into a single entity, enhancing strength and centralizing functions[21].

Eric B. Duoss et al.[22] comprehensively formulated the concept and theory of cellular fluidics in 2021. Previously, Hiroki Yasuga et al.[23] conducted similar work but did not offer a comprehensive theory from the perspective of cellular fluids.

Inspired by the multinucleate structure of syncytia, we have introduced the concept of cellular fluidics into stirred reactors, utilizing 3D printing technology to design and manufacture structured cellular reactors (Fig. 1a). These cellular reactors feature (1) catalyst components securely bound within the cell units, preventing leakage during the reaction and facilitating easy separation afterward. (2) highly customizable reactor structure, enabling precise spatial distribution of catalytic and oxidative components in the reaction system and facilitating adequate interfacial contact with the substrate; and (3) the internal connectivity of cellular reactors facilitates passive diffusion of liquids within, affording opportunities for interface renewal. This cellular reactor generates a stable water-oil interface, significantly increasing the contact area between the two phases and enhancing mass transfer flux. Additionally, it effectively confines $H_2SO_4$, $K_2CO_3$, $H_2O_2$, and $H_5IO_6$ within the aqueous phase inside the reactor, allowing precise spatial control in the reaction system. The substrates are dissolved in the organic continuous phase outside the reactor. Upon rotation, the stirred structured cellular reactor further enhances mass transfer, simultaneously eliminating external diffusion and maintaining interfacial stability. Moreover, the reactor also inherits the easy separation characteristic of structured catalysts, relying merely on the physical property differences between water and oil phases to achieve separation and recovery of reaction components and substrates effortlessly.

## Results and discussion

Design of cellular reactors. Inspired by syncytia, cellular reactors comprise individual cellular structural units, each capable of storing liquid and constructing a liquid-liquid interface. Units are interconnected through digital structural design, allowing liquids to transfer between units, refreshing interface reactants, and enabling multicellular cooperative functions. Consequently, the design of individual cellular units significantly impacts cellular reactor construction. These units must possess sufficient porosity to securely retain substantial amounts of liquid while constituting a spatial network with robust mechanical stability and the ability to withstand the loaded liquid's mass without breakage during movement. In this study, we adopt the standard and quickly processed cubic lattice as the fundamental unit structure of the cellular reactor, forming the basis for the entire reactor. Eukaryotic cells naturally contain a series of network structure proteins called the cytoskeleton, serving crucial roles in providing mechanical support, maintaining cell shape, and resisting deformation[24–26]. In the cellular reactor, the framework structure functions analogously, providing mechanical support, ensuring interface stability, and resisting deformation, resembling the physiological function of the cytoskeleton.

The cell, being the fundamental unit of the reactor, significantly influences its conformation. Four dimensionless numbers were defined to describe the cell's liquid storage capacity and ability to construct a liquid-liquid interface, quantifying the cellular unit structure's impact on the reaction process. Among these, the characteristic diameter $D^*$ is calculated as follows:

$$D^* = \frac{D}{L} \tag{1}$$

where $D$ represents the diameter of the cell truss, $L$ is the cell length (Classic length 2 mm), and $D^*$ is a characteristic dimension of the cell. Porosity, $\varepsilon$ is another dimensionless number that reflects the percentage of cells' capacity for holding liquid. $A_{SL1}^* = \frac{A_{SL1}}{L^2}$, and $A_{LL}^* = \frac{A_{LL}}{L^2}$ are dimensionless numbers relating the solid-liquid1 and the liquid-liquid contact areas, respectively. All symbols used in this study are listed in Table 1.

**Table 1 | Symbols used in this study**

| Symbol | Physical meaning | Clarification |
|---|---|---|
| L1 | Liquid adsorbed by the cellular reactor | Usually an aqueous solution carrying catalytic components |
| L2 | Liquid outside the cellular reactor | Usually an organic phase that carries the substrate |
| S | Solid trusses supporting cellular reactor | Manufactured by 3D printing to support, attach and construct phase interfaces |
| L | Length of a cell side (m) | Physical quantities that describe the size of a single cell |
| D | Diameter of the cell truss (m) | Physical quantities describing cell truss dimensions |
| $D^*$ | Characteristic diameters of cells | $D^* = D/L$, dimensionless number |
| $A_{xy}$ | Contact area of phases x and y (m$^2$) | x, y represent two phases in arbitrary contact |
| $\sigma_{xy}$ | Interfacial tension between the x and y phases (N/m) | |
| $\rho$ | density (g/cm$^3$) | |
| $\Delta\rho_{xy}$ | Difference in density between phases x and y (g/cm$^3$) | |
| n | Cell freedom, number of liquid-liquid interfaces | Dimensionless number |
| N | Total degrees of freedom of the cellular reactor | The sum of degrees of freedom of each single cell, a dimensionless number |
| g | Gravitational acceleration (m/s$^2$) | |
| $\theta$ | Contact angle (°) | |
| $\lambda$ | Equivalent wetting factor | A dimensionless measure of cell wetting area |
| $\varepsilon$ | Porosity | A dimensionless number that reflects the percentage of cells' capacity for holding liquid |
| s | wetted perimeter (m) | Perimeter of contact between solid and liquid |

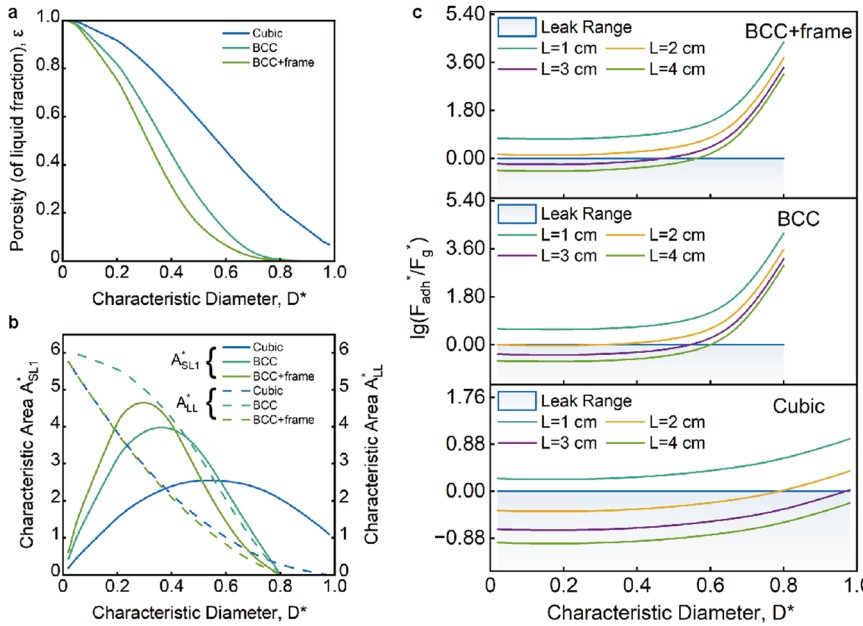

**Fig. 2 | The structural characteristics of the cell units include porosity, solid-liquid and liquid-liquid contact areas, and the characteristic magnitudes of adhesion and gravity. a** Porosity $\varepsilon$ as a function of the characteristic diameter $D^*$, reflecting the liquid-holding capacity of a given cell. **b** Implicit function of the characterized solid-liquid 1 interface area and the liquid 1-liquid 2 interface area versus $D^*$, reflecting the ability of a given cell to construct an interface. **c** The relative characteristic capillary force versus the characteristic gravitational force magnitude reflects how well the cell holds liquid (it is worth noting that for BCC and BCC + frame, porosity $\varepsilon$ exists when $D^* < 0.8$ for practical purposes). Source data are provided as a Source Data file.

The paper investigates three common types of cubic cells (Fig. 1b) known for constructing stable lattice systems: simple cubic (Cubic), body-centered cubic without frame (BCC), and body-centered cubic with frame (BCC+frame). Figure 1c illustrates various parameters of cellular operation using the Cubic cell unit as an example.

As $D^*$ increases from 0 to 1, the porosity of different cubic cell structures decreases, resulting in a reduced fluid holding capacity (Fig. 2a) (refer to the Source). The implicit functions $A^*_{SL1}$ and $A^*_{LL}$ are dependent on the cell's characteristic diameter $D^*$. $A^*_{SL1}$ decreases to varying degrees with the increase of $D^*$, whereas $A^*_{LL}$ initially increases and then decreases with the increase of $D^*$ (Fig. 2b). By selecting the optimal $D^*$, it becomes possible to obtain suitable solid-liquid1 interfacial area and liquid 1-liquid 2 interfacial area, ensuring both the best liquid holding effect and complete contact between the water-oil two-phase liquid reactants.

The stability and spontaneity of liquid retention in a cellular reactor, acting as a carrier of liquid, rely on the ability of cell trusses to generate capillary forces that surpass the force of gravity on the liquid. The expressions for capillary force and gravity in a closed capillary are

represented as follows:

$$F_{adh} = \sigma \cos \theta \int ds = \sigma \cos \theta 2\pi r \qquad (2)$$

$$F_g = \rho g h \pi r^2 \qquad (3)$$

where $\sigma$ denotes the surface tension of the liquid, $r$ is the radius of the capillary, $\rho$ is the density of the liquid, $g$ is the acceleration due to gravity, and $h$ is the height of the capillary.

When the forces of gravity and capillarity are equal, the system reaches equilibrium, maximizing the liquid surface height in the capillary tube. However, the cellular reactor operates as an open system, with intermittent contact between the liquid and the solid wall within the same dimension. In the vertical direction, the cross-sectional perimeter of the truss varies with the height, allowing the area of the solid-liquid interface to be expressed as a function of the height $h$ [$s(h)$, $s$ is the wetting perimeter]. Consequently, the formula for capillary force and gravity in the cellular unit becomes:

$$\langle \bar{F}_{adh} \rangle_{xy} = \sigma \cos \theta \int s(h) ds \qquad (4)$$

$$F_g = \rho g L^3 \varepsilon \qquad (5)$$

$\theta$ denotes the three-phase contact angle between liquid 1 and 2 and the solid. Gravity is a volumetric force, dividing both capillary force and gravity by the cell volume $L^3$ to eliminate intercellular differences due to size:

$$F_{adh}^* = \frac{\sigma \cos \theta}{L^3}(s) \qquad (6)$$

$$F_g^* = \rho g \varepsilon \qquad (7)$$

The characteristic gravity curves remain identical for cells with the same spatial configuration but different side lengths. It occurs because the volume factor, attributed to the side length, is eliminated after normalization. Figure 2c shows the mathematical relationship between $\lg\left(\frac{F_{adh}^*}{F_g^*}\right)$ and $D^*$. Consequently, for any given $D^*$, if the characteristic capillary force exceeds the characteristic gravity force, the cellular reactor can retain liquid. However, considering the limitation of the capillary height[27] $h = \sqrt{\frac{\sigma_{xy}}{\Delta \rho_{xy} g}}$ (where $\sigma_{xy}$ represents the interfacial tension of the two phases x and y, and $\Delta \rho_{xy}$ indicates the difference in density between the two phases), When the characteristic size $L$ of a cell exceeds $h$, the honeycomb reactor loses its liquid-holding capacity, regardless of the cell unit's configuration. For instance, in the case of water and cyclohexane, the capillary size reaches up to 4.5 mm.

Various post-treatments are employed to modify the surface energy and wettability of the cell truss to enhance the versatility of cellular reactors in diverse reaction systems, such as gas-liquid and liquid-liquid systems. For instance, after oxygen plasma cleaning, the truss surface exhibits superhydrophilicity, promoting stable water adsorption within the cellular reactor in air and oil environments. Conversely, plasma cleaning with perfluorosilane grafting treatment results in a superhydrophobic truss surface, facilitating stable adsorption and maintenance of the oil phase in air and water (Supplementary Fig. 1). Cellular reactors were prepared based on these conditions tailored for different internal and external phase systems. As air, water, and cyclohexane are colorless, suitable dyes were added to visualize the solvents. Supplementary Fig. 2 depicts the reactor holding water and cyclohexane in air, cyclohexane in water, and water in cyclohexane. Regardless of whether the liquid in the reactor was an

aqueous or an oil-phase solvent, the liquid remained stable with no leaks, confirming the cellular reactor's excellent liquid-carrying capability. Furthermore, for the cellular reactor's ability to absorb, transfer, and retain liquids, please refer to Supplementary Movies 1, 2, and 3.

Working principle of cellular reactors. In the initial state (Fig. 3a), the inside and outside of the cellular reactor are filled with air. When the cellular reactor contacts liquid 1, the trusses' remarkable affinity for liquid 1 drives the liquid into the cellular reactor, displacing the original air inside the cell, while the external phase remains air (stage 1). The cellular reactor securely holds the internal phase liquid, thanks to the capillary forces prevailing over the force of gravity on the liquid.

The cellular reactor's ability to adsorb liquid results from capillary forces prevailing over the liquid's gravitational force, while retaining the fluid in another phase occurs due to a combination of solid-liquid and liquid-liquid interfacial tensions.

When the cellular reactor containing liquid 1 is immersed in liquid 2 (phase 3), liquid 1 remains within the cell if the cellular truss exhibits a higher affinity for liquid 1 than liquid 2. For detailed calculation steps, please refer to the supporting information (Supplementary Figs. 3 and 4, Supplementary Tables 1-3).

$$A_{LL} - A_{SL1} \cos \theta < 0 \qquad (8)$$

where $A_{LL}$ is the contact area of liquids 1 and 2, $A_{SL1}$ is the contact area between solid and liquid 1, and $\cos \theta$ is the contact angle between liquid 1 and 2 and solid three-phase.

Considering the cellular reactor as a whole, mutual coordination between cells is necessary, which inevitably causes changes in $A_{LL}$ and $A_{SL1}$ of neighboring cells (Fig. 3b). $A_{LL}$ and $A_{SL1}$ are calculated according to the following equations, respectively:

$$A_{LL} = nL^2(1 - D^*)^2 \qquad (9a)$$

$$A_{LL} = nL^2\left[1 - \frac{\pi}{4}(D^*)^2\right] \qquad (9b)$$

$$A_{SL1} = \pi L^2 D^*(1 - D^*)\lambda \qquad (10)$$

where $n$ represents the degree of cell freedom, indicating the number of contact surfaces of liquids 1 and 2 within each cell; $L$ stands for the length of the cell unit; $D^*$ denotes the characteristic diameter of the truss, and $\lambda$ is a dimensionless number describing the wetting properties of the cell. Equation 9a applies to SC and BCC+frame, while Eq. 9b applies to BCC.

To establish the range of characteristic cell diameters, Eqs. 8, 9a and 10 were simultaneously solved, yielding the following relationship:

$$\left(\pi \frac{\lambda}{n} \cos \theta + 1\right)^{-1} < D^* < 1 \qquad (11)$$

Among these factors, the contact angle is determined by the wettability of liquid 1, 2, and the solid surface, and it can be modified through surface treatment. The truss structure of the cell dictates that $D^* < 1$. The equivalent wetting coefficient $\lambda$ is calculated through quality analysis of cell units using 3D modeling software and is implicitly related to the characteristic diameter $D^*$ (Fig. 3c). Moreover, the lower limit of the value range $D_{\min}^*$ for the characteristic diameter depends on both the equivalent wetting coefficient $\lambda$ and the degree of freedom $n$, thus forming a function of the characteristic diameter $D^*$. The relationship between $D_{\min}^*$ and $D^*$ is illustrated in Fig. 3d. Since $D_{\min}^* < D^*$, the cells, where $D_{\min}^*$ falls below the straight line $y = x$, exhibits stable liquid-holding capabilities. Plot the Cubic and BCC+frame curves in Figure 3d according to Equation 11. Furthermore, combine Equations 8, 9b, and 10 and plot the resulting BCC curve in Figure 3d. Cellular

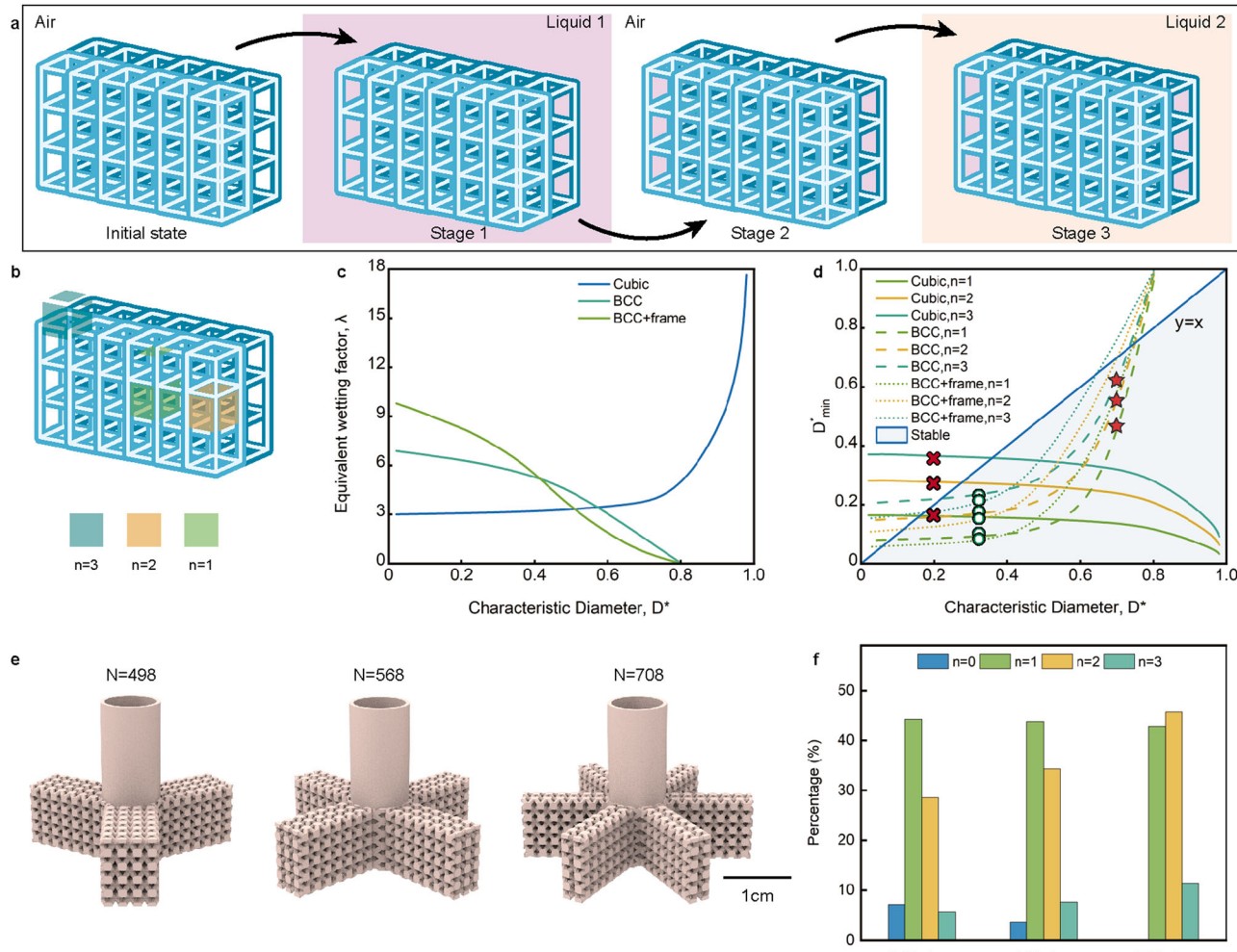

**Fig. 3 | The working performance of the cellular reactor. a** The ability of the reactor to spontaneously adsorb a liquid and retain it in another incompatible liquid. **b** variations in the degrees of freedom of cells at different locations within the reactor. **c** the equivalent wettability coefficient λ as a function of $D^*$. **d** the lower limit of the range of values for the equivalent diameter, $D_{min}^*$, as a function of $D^*$

(asterisks denote the working points used throughout the study, circles represent liquid retention, and crosses indicate liquid leakage). **e** diverse branching configurations of the cellular reactor. **f** the distribution of degrees of freedom in the differently branching cellular reactors. Source data are provided as a Source Data file.

reactors with varying unit sizes and characteristic dimensions were designed, manufactured, and subjected to liquid-holding experiments. The principle of liquid retention in cellular reactor was validated (see Supplementary Fig. 5, Supplementary Table 5).

The spatial configuration, size characteristics, and overall structure of the cellular reactor significantly influence its catalytic efficiency. The total degrees of freedom vary among cellular reactors with the same number of cells but different numbers of paddle blades (3, 4, and 6 blades, respectively–Fig. 3e, f). The total degree of freedom (N) is the sum of all cell degrees of freedom, given by $N = \sum_i n_i$. For the three cellular reactors tested, which comprised 420 cells, the experimental results revealed that the reactor with 6 blades exhibited the fastest reaction rate during the addition reaction of dihydropyran and alcohol (Supplementary Table 6). This disparity in reaction rates is attributed to the differences in the contact area between liquid 1 and liquid 2, resulting from varying degrees of freedom.

Chromogenic reaction. To visualize interphase mass transfer behavior, colored compounds were employed in the experiment. The process was conducted with the cellular reactor remaining stationary to facilitate the capture of color changes in the cellular reactor. The chromogenic reagent bromothymol blue (BTB) was dissolved in cyclohexane as the external phase, yielding a transparent light yellow solution (Fig. 4a). The internal phase of the cellular reactor consisted of

an aqueous NaOH solution. BTB molecules diffuse from the bulk phase to the oil-water interface, driven by concentration gradients. If BTB molecules reach the interface, they react with NaOH to form sodium bromothymol blue (SBB) upon contact between the two phases. Conversely, BTB molecules will return to the oil phase. In the water-cyclohexane system, SBB readily dissolved in the water phase, resulting in an immediate blue coloration of the water phase. The amount of blue solution within the cellular reactor gradually increased with prolonged reaction time. Notably, no leakage of the blue liquid into the oil phase occurred, and the color of the oil phase remained unchanged. This observation confirms that the reaction occurred exclusively at the interface, with the oil-soluble reactant transforming into a water-soluble product and crossing the water-oil interface to enter the other phase. Please refer to Supplementary Movie 4 for more details.

In a control experiment, the colorimetric reaction of Neutral Red (NR) with sodium hydroxide was also conducted (Fig. 4b). NR molecules exist in the form of hydrochloride salts. Their color change ranges from pH 6.4 to 8, turning red to yellow. Initially, NR dissolved in ethyl acetate appears pale pink, with 1 M NaOH as the aqueous phase. As stirring proceeds, the solution's pale pink color gradually shifts to pink-orange and light yellow. This experimental observation indicates that NR molecules undergo a chemical reaction at the interface with NaOH, where NaOH neutralizes the HCl group and successfully returns

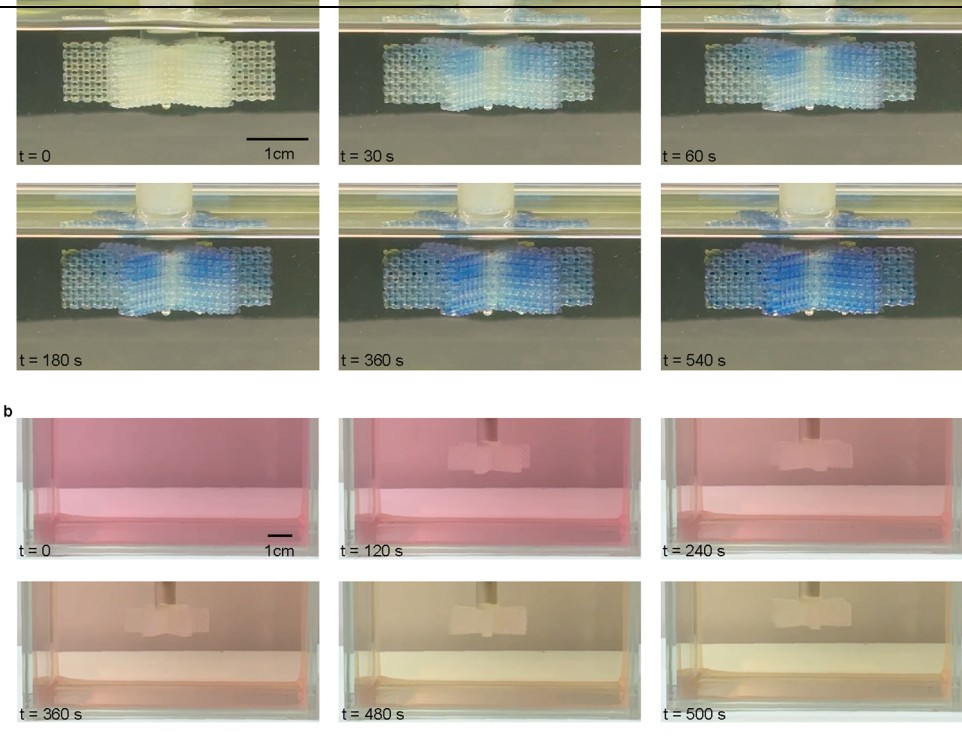

**Fig. 4 | Chromogenic reaction. a** Colorimetric reaction of bromothymol blue (soluble in cyclohexane) with sodium hydroxide (aqueous solution). **b** Colorimetric reaction of neutral red (soluble in ethyl acetate) with sodium hydroxide (aqueous solution).

## Table 2 | Addition reactions of alcohols to 3,4 −2H-dihydropyran

| Entry | R−OH | Time 1 | Conversion rate 1 | Time 2 | Conversion rate 2 |
|---|---|---|---|---|---|
| 1 | | 60 min | 53.3% | 300 min | 99.2% |
| 2 | | 60 min | 47.3% | 300 min | 97.6% |
| 3 | | 60 min | 37.0% | 300 min | 97.0% |
| 4 | | 60 min | 38.7% | 300 min | 97.7% |

Please refer to Supplementary Figs. 9–12 for the GC-MS spectra of entries 1–4.

to the organic phase with a color change. At the reaction's end, the sodium hydroxide cellular reactor can be pulled upwards from the container to achieve phase separation. For the colorimetric reaction of NR and the post-reaction separation process, refer to the video in Supplementary Movie 5. Additionally, models for interface reactions have been proposed; please refer to Supplementary Figs. 6 and 7.

To assess the effectiveness of cellular reactors in practical reactions, we evaluated four distinct types of liquid-liquid reactions. These reactions included $H_2SO_4$-catalyzed addition reactions, $K_2CO_3$-catalyzed condensation reactions, coupling reactions facilitated by hydrogen peroxide and periodate as oxidizing agents, and oxidation of thioethers. These reactions share common characteristics, including the involvement of acids, bases, or oxidants, the need for vigorous stirring to promote mixing, and the requirement for quenching residual corrosive or oxidative substances, demulsification, and oil-water separation post-reaction, making the process somewhat cumbersome.

Please refer to Supplementary Fig. 8 for the overall schematic diagram of the reaction setup.

Addition reaction of alcohols to 3,4-2H-dihydropyran. Hydroxyl protection plays a crucial role in organic synthesis, and dihydropyran serves as an effective hydroxyl-protecting reagent for acid-catalyzed addition reactions with alcohols[28], leading to the formation of tetrahydropyran ethers. In this study, the substrate dihydropyran and the alcohol were dissolved in cyclohexane as the outer phase, while dilute sulfuric acid catalyzed in the inner phase. The reaction of benzyl alcohol with dihydropyran demonstrated an impressive 99.2% conversion of dihydropyran and nearly 100% selectivity (Table 2). Subsequently, several other alcohols were employed as substrates, and the addition reactions of dihydropyrans with cyclohexanol, n-hexanol, and n-octanol all achieved over 95% conversion.

Knoevenagel condensation reaction. The alkali-catalyzed Knoevenagel condensation reactions are widely employed for carbon chain

**Table 3 | Knoevenagel condensation reactions of benzaldehyde and malononitrile**

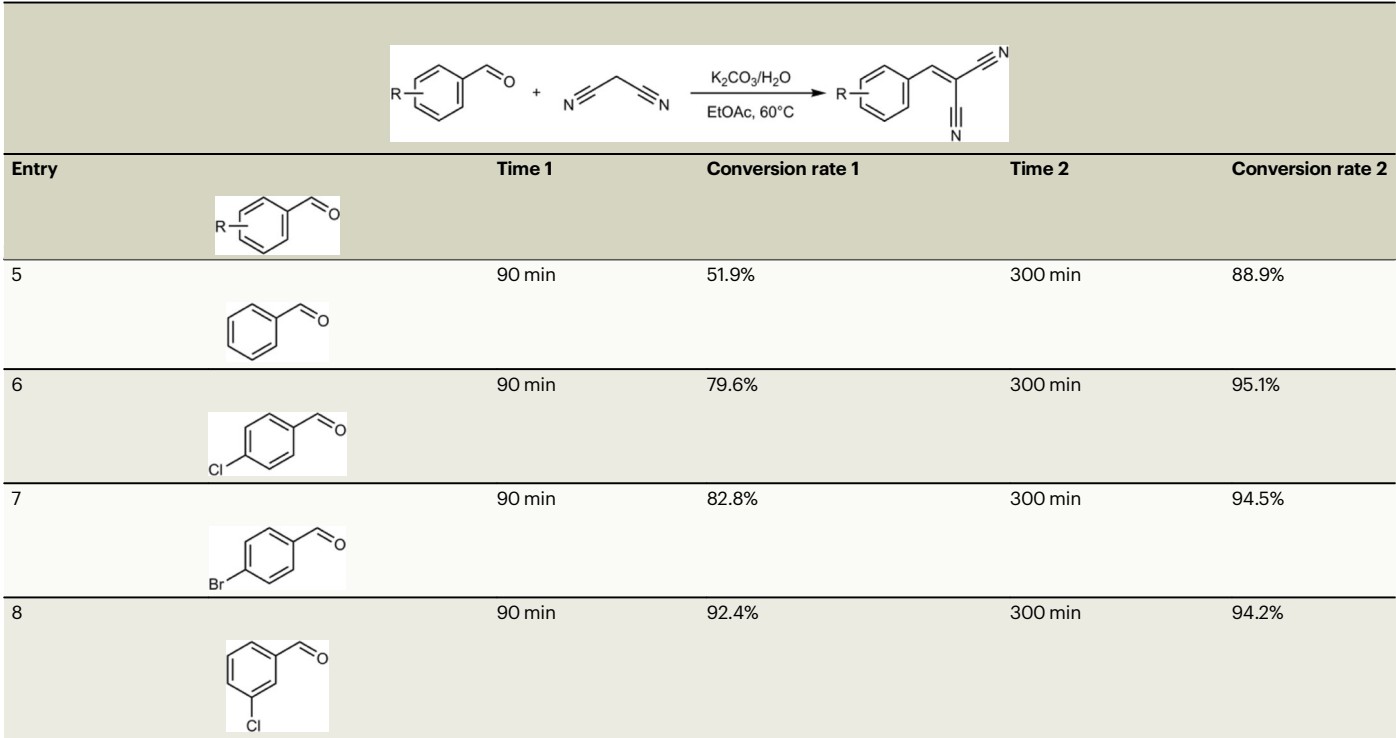

| Entry | | Time 1 | Conversion rate 1 | Time 2 | Conversion rate 2 |
|---|---|---|---|---|---|
| 5 | | 90 min | 51.9% | 300 min | 88.9% |
| 6 | | 90 min | 79.6% | 300 min | 95.1% |
| 7 | | 90 min | 82.8% | 300 min | 94.5% |
| 8 | | 90 min | 92.4% | 300 min | 94.2% |

Please refer to Supplementary Figs. 13–17 for the GC-MS and NMR spectra of entries 5–8.

elongation in organic synthesis[29]. Reactive methylene-containing compounds react with aldehydes or ketones, leading to the formation of α,β-unsaturated compounds and the extension of conjugated systems. In this study, the external phase consisted of an ethyl acetate solution containing malononitrile and benzaldehyde, while the internal phase served as the catalyst, comprising $K_2CO_3$ dissolved in water. The reaction between 4-chlorobenzaldehyde and malononitrile demonstrated a 95.1% conversion of 4-chlorobenzaldehyde within 300 min (Table 3). Similarly, the conversion rates for benzaldehyde, 4-bromobenzaldehyde, and 3-chlorobenzaldehyde reached 89%, 95%, and 94%, respectively. Blank experiments confirmed that cellular reactors without adsorbed aqueous alkaline-containing solutions failed to catalyze the reaction (Supplementary Table 7), ensuring that the cellular reactor-loaded aqueous $K_2CO_3$ solution plays the catalytic role.

Coupling reactions of sulfhydryl compounds. The disulfide bond is a crucial functional group in various natural small molecules and proteins[30]. Many biologically active small molecules contain disulfide bonds, making them common components in pharmaceutical compounds[31]. Examples include Vapreotide Acetate[32] (an antitumor drug), Eptifibatide[33] (a cardiovascular drug), and Lipoic Acid (a neurological drug). Synthesizing sulfur-containing dimers involves the oxidation of sulfhydryl compounds to form disulfide bonds. In this study, hydrogen peroxide, a green oxidizing agent, served as the internal phase, while mercapto compounds were dissolved in ethyl acetate as the external phase. Remarkably, hexanethiol, phenylethanethiol, 4-fluorobenzenethiol, and 4-methylbenzenethiol all achieved nearly 100% conversion and selectivity within just 30 min, (Table 4) demonstrating the effectiveness of the reaction.

Oxidation reactions of sulfur ethers. The oxidation of sulfur ethers allows the synthesis of sulfoxide and sulfone[34], common structures found in pharmaceutical intermediates[35]. Cellular reactors play a crucial role in stabilizing the phase interface, enabling prolonged contact

**Table 4 | Coupling reactions of sulfhydryl compounds**

| Entry | R—SH | Time | Conversion rate[a] |
|---|---|---|---|
| 9 | (hexanethiol) | 30 min | ~100% |
| 10 | (phenylethanethiol) | 30 min | ~100% |
| 11 | (4-methylbenzenethiol) | 30 min | ~100% |
| 12 | (4-fluorobenzenethiol) | 30 min | ~100% |

[a]The substrate concentration was analyzed using GC-MS, and at 30 min, it was found to be below the detection limit, indicating complete transformation. Please refer to Supplementary Figs. 18–24 for the GC-MS and NMR spectra of entries 9–12.

between the sulfur ether and the oxidizing agent, resulting in one-step oxidation to sulfone. Reaction 14 tests demonstrated that within 48 h, all benzyl sulfide was converted to a sulfone with a selectivity close to 100%, and no production of sulfoxide was detected. (Table 5) Periodic acid, with its high oxidation potential $[E_{MF} (H_5IO_6/IO_3^-) = 1.60 \text{ V}]$、$[E_{MF} (IO_3^-/I_2) = 1.209 \text{ V}]$, proved to be a suitable oxidant for this reaction. The product molecules dissolved in ethyl acetate, while the reduction products (iodic acid, hypoiodic acid, and hydriodic acid) were soluble in the aqueous phase, allowing for easy separation from the products using the cellular reactor.

**Table 5 | Reactions for the oxidation of thioethers to sulfones**

$$R^1-\underset{S}{S}-R^2 \xrightarrow[\text{EtOAc, rt}]{H_5IO_6/H_2O} R^1-\underset{O}{\overset{O}{S}}-R^2$$

| Entry | $R^1-\underset{S}{S}-R^2$ | Time 1 | Conversion rate 1[a] | Time 2 | Conversion rate 2[b] |
|---|---|---|---|---|---|
| 13[b] | | 70 min | ~100% | - | - |
| 14 | | 24 h | 24.3% | 48 h | 99.9% |
| 15 | | 30 min | 71.3% | 4 h | ~100% |
| 16 | | 30 min | 30.0% | 6 h | ~100% |
| 17[c] | | 24 h | 23.5% | 36 h | ~100% |

[a]The substrate concentration was analyzed using GC-MS, and at 30 min, it was found to be below the detection limit, indicating complete transformation. Reaction 17 was analyzed by UPLC-MS. Please refer to Supplementary Figs. 25–33 for the GC-MS and NMR spectra of entries 13–17. 13 [b]: The oxidation product is sulphoxide, 17 [c] The oxidation product is sulphoxide.

With the participation of the cellular reactor, all four types of reactions mentioned above yield favorable results, and subsequent separation becomes remarkably simple, requiring only the lifting of the reactor out of the system.

The cellular reactor facilitates easy regeneration and replacement of new liquid components, which can be accomplished by washing with a dilute ethanol solution, drying, and re-adsorbing the liquid. The reactor was used to repeatedly perform the acid-catalyzed addition reaction (Entry 1) five times without any decrease in reaction yield (Supplementary Table 8). No significant damage was observed visually or in SEM images (Supplementary Fig. 34), indicating the cellular reactor's high reusability and stability. It further reduces the application costs and enhances its practical prospects. Two other resin materials were also used to fabricate cellular reactors (Supplementary Table 4), achieving similar effects and broadening the material range (Supplementary Tables 9 and 10). Moreover, the cellular reactor material demonstrated good tolerance to all reaction components mentioned above (Supplementary Figs. 35–38), and have good mechanical stability (Supplementary Fig. 39). The exceptional catalytic performance of the cellular reactor is not only the result of its structured design but also due to its ability to create favorable fluid dynamics. Compared to two types of commercially available generic stirring blades, the cellular reactor offers significant catalytic performance advantages (Supplementary Table 11) and accompanies good flow field disturbance (Supplementary Figs. 40 and 41, Supplementary Tables 12 and 13). Furthermore, the necessity of rotation has been confirmed. In fact, a very low rotational speed (50 rpm) is sufficient to significantly enhance the chemical reaction rate (Supplementary Table 14). The simulation of solid precipitation experiments yielded positive outcomes (Supplementary Table 15), indicating that solid impurities not participating in chemical reactions do not adversely affect the performance and efficiency of the cellular reactor. Cross-contamination experiments demonstrate that for biphasic systems with slight mutual solubility, using cellular reactors does not introduce additional levels of contamination (Supplementary Tables 16 and 17, Supplementary Figs. 40–43). Compared to advanced chemical reactors and extraction tools (Supplementary Table 19), cellular reactors exhibit comparable operational efficiency but have fewer steps and shorter operation times.

Inspired by the syncytium structure in organisms and the construction of biological membranes, we have developed a cellular reactor for liquid-liquid reaction systems. This reactor has dual functions of enhancing interphase contact and restricting the spatial distribution of reaction components, achieving high synthesis yields in liquid-liquid reactions, and improving the safety and separability of reactions. It allows for precise control over the spatial distribution of single-phase components in liquid-liquid reactions, ensuring catalytic efficiency. The reactor applies to completely immiscible and partially miscible water-oil biphasic systems, capable of performing acid-catalyzed addition reactions, base-catalyzed condensation reactions, and oxidation reactions involving strong oxidants. Most reactions can achieve conversion rates > 95% within a short period. Chemical reactions in this cellular reactor are characterized by their simplicity and efficiency, straightforward separation of catalyst components and products, enhanced safety, good reproducibility, and reusability. It provides a venue and design paradigm for liquid-liquid reaction systems.

## Methods

### Fabrication of cellular reactors

The cellular reactors were designed using 3D drawing software and fabricated with a light-curing 3D printer. Unless otherwise specified, this study's cell sizes (L) were 2 mm and belonged to the type BCC (no frame). For more parameters, please refer to the supporting Supplementary Table 3. After 3D printing, the cellular reactors were cleaned with isopropyl alcohol to remove excess resin and air-dried. The reactors were transferred to a post-curing device to complete the curing process and treated at 50 °C for 300 min. The 3D printer used was Form 2, and the post-curing device was Form Cure, both from Formlabs. Commercial Clear Resin (FLGPCL04), Tough 2000 Resin (v1 RS-F2-TO20-01), and High Temp Resin (v2 RS-F2-HTAM-02) were employed for the printing process. All types of cellular reactors referred to in the main text are categorized as Reactor-C in Supplementary Table 4.

### Surface modification

The cellular reactor employed two distinct surface modifications to cater to different reaction systems. The reactors were subjected to high-power oxygen plasma cleaning for 20 min to render the surface hydrophilic. For hydrophobic reactors, a 1H,1H,2H,2H-perfluorodecyltrimethoxysilane chemical vapor deposition at 90 °C was performed for 3 h. The wettability can be adjusted by varying the treatment time.

### Characterization

Contact angles and interfacial tension data were obtained on a commercial contact angle system (OCA50, Dataphysics, Germany). All tests are performed at ambient temperature unless otherwise stated. The 1H NMR spectra were recorded on a Bruker Avance II 400 instrument. All 1H NMR experiments are reported in $\delta$ units, parts per million (ppm), and were measured relative to the signals for residual chloroform (7.26 ppm) in the deuterated solvents. Data for 1H NMR are reported as follows: chemical shift ($\delta$ ppm), multiplicity (s = singlet, d = doublet, t = triplet, q = quartet, p = quintet, m = multiplet, dd = doublet of doublets, dt = doublet of triplets…etc, br = broad), coupling constant (Hz) and integration. The total organic carbon (TOC) data was obtained using a commercial TOC analyzer (multi N/C 2100 S, Analytik Jena, Germany) in a 300 μL solution. The authors custom-built the

particle image velocimetry system comprising a laser (532 nm), high-speed camera, synchronizer, and computer.

## Colorimetric reaction

A solution was prepared by dissolving 0.2 g of bromothymol blue in 500 mL of cyclohexane, which was then stirred for 30 min until a clear light yellow solution was obtained. The solution was filtered using a 0.22 μm filter membrane to remove undissolved particles. The cellular reactor was adsorbed with a 0.1 M NaOH solution. Then, the entire cellular reactor was immersed in cyclohexane and left standing while observing the color changes of the cellular reactor and the organic phase separately.

The solution is filtered through a 0.22 μm filter membrane to remove any potentially undissolved particles. A 0.1 M sodium hydroxide solution is adsorbed into the cellular reactor. Subsequently, the cellular reactor is fully immersed in ethyl acetate, stirring at 50 rpm to observe color changes in the organic phase.

## Addition reaction of alcohols to 3,4-2H-dihydropyran

For the typical addition reaction of dihydropyrans with alcohols, 12.5 mmol of dihydropyran and 12.5 mmol of benzyl alcohol (1.0 equiv) were combined in 50 mL of cyclohexane. The cellular reactor was adsorbed with 1 M $H_2SO_4$ and placed into the organic phase, stirring at 50 rpm. The reaction was conducted in a custom-made three-neck flask heated at 60 °C. Various cellular reactor structures were employed in the experiments to validate their efficacy. Extension experiments used various alcohols as substrates, specifically n-octanol, n-hexanol, and cyclohexanol.

## Knoevenagel condensation reaction

For the typical Knoevenagel condensation reaction, 4-chlorobenzaldehyde (2.5 mmol) and malononitrile (7.5 mmol, 3.0 equiv) were combined in 50 mL of ethyl acetate. The cellular reactor was adsorbed with 40 mM $K_2CO_3$ and placed into the organic phase, stirring at 50 rpm. The reaction was conducted in a custom-made three-neck flask heated at 60 °C. Various cellular reactor structures were employed in the experiments to validate their efficacy. Extension experiments used various alcohols as substrates, specifically benzaldehyde, 4-bromobenzaldehyde, and 3-chlorobenzaldehyde.

## Coupling reactions of sulfhydryl compounds

In a typical reaction for generating disulfide bonds, 5 mmol of the mercaptan compound was dissolved in 50 mL of ethyl acetate, and 0.1 equivalents of sodium iodide were added as an auxiliary. A 15 mmol (3 equivalents) dilute aqueous hydrogen peroxide solution was adsorbed into the cellular reactor and introduced into the organic phase as an oxidant. The reaction occurred in a custom-made three-neck flask, catalyzed at 50 rpm at room temperature. Thiol compounds were used, including hexanethiol, phenylethanethiol, 4-fluorobenzenethiol, and 4-methylbenzenethiol.

## The oxidation reaction of thioethers

In a typical oxidation reaction of sulfur ethers to sulfoxides, 1 mmol of sulfur ether was dissolved in 50 mL of ethyl acetate, and 5 mmol (5 equiv.) of periodate dissolved in water was adsorbed into the cellular reactor as an oxidizing agent, and introduced into the organic phase. The reaction was conducted in a custom-made three-neck flask and catalyzed at 50 rpm at room temperature. Thioether compounds include benzyl thioether, phenyl thioether, tetrahydrothiophene, allyloxyphenylthiol, and 4-nitrobenzyl thioether. The progress of the reaction was monitored using GC-MS/UPLC-MS.

## Recycling and regeneration experiment

In the cyclic regeneration experiments, we selected a typical addition reaction of dihydropyran with benzyl alcohol as the probe reaction.

After the first catalytic reaction, the cellular reactor was immersed in a 10% ethanol solution and cleaned ultrasonically for 1 min, repeating the process three times. It was then thoroughly rinsed with water and dried. For the second use, the plasma cleaning step was repeated, and the regeneration was accomplished by adsorbing the $H_2SO_4$ solution before proceeding to the next experiment. This regeneration process was repeated four times. The progress of the above four kinds of reactions was all monitored using GC-MS.

## Reporting summary

Further information on research design is available in the Nature Portfolio Reporting Summary linked to this article.

## Data availability

The data for this study are available in this article or are included in the Supplementary Information, which is also available from the corresponding authors upon request. Source data are provided with this paper.

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

## Acknowledgements

We gratefully acknowledge the support of this research by the National Natural Science Foundation of China (No. 22372025, S.T.), the Fundamental Research Funds for the Central Universities (No. DUT22LAB607, J.C. and S.T., DUT22QN226, J.C. and S.T.) and Project 1912 Funds, S.T.

## Author contributions

S.T. provided manuscript revision and experimental guidance. All other work was done by J.C.

## Competing interests

The authors declare no competing interests.
