## [Peer Review File · Nature Communications]

Liquid-Liquid Reactions Performed by Cellular ReactorsREVIEWER COMMENTS

Reviewer #1 (Remarks to the Author):

In this paper, Jinzhe Cao et. al. reported an a biomimetic 3D-printed cellular reactor for organic reactions such as acetal formation reaction between 3,4 -2H-dihydropyran and primary alcohols under acidic conditions, Knoevenagel condensation reaction under basic conditions, oxidation reactions of thiols by hydroperoxide and oxidation reactions on thiol ethers by HIO₄. The authors described the design and manufacture of the cellular reactors by 3D-printing, however, however, it is very hard to judge whether the new reactors possessed some enhancements for selected organic transformations compared with conventional reactors. It is suggested major revision and resent for review. Some other points need to be addressed as following:

(1) The cellular reactors in this paper are made of resin FLGPCL04 and some treatments need done to make them hydrophilic and hydrophobic. Is the resin stable enough when it is contacted with hot organic solvents under acidic, basic and oxidative conditions? It is worth investigating if the cellular reactors remained the same after several cycles of reactions. And any other materials than resin FLGPCL04 are suitable for 3D printing and the selected reactions?

(2) In this paper, GC-MS was used to monitor the reaction process. However, that is not enough. HPCL and NMR are highly suggested, in addition to GC-MS, especially for some compounds which are not volatile.

(3) It is very hard to judge whether the new reactors possessed some enhancements for selected organic transformations compared with conventional reactors. Some metrics need to be proposed to evaluate this point, other than conversions and yields of the well-developed organic reactions.

(4) For the oxidation of thioethers, not only to see the conversions of substrates, the confirmation of the products is very crucial. Sulfones and sulfoxides are both possibly obtained? Please give more detailed information.

(5) The title actually can not show what has been done in the paper. Some more connections between the newly constructed reactors and the well-studied organic reactions are suggested to be strengthened. At this moment, the two parts of work are just put together, as many more reactions could be taken and tested, but why just the four reactions were tested by this new reactor.

(6) The liquid-liquid reactions are complex beside the simple phase separations in fact. Take the water and EtOAc as example, in aqueous phase, EtOAc was somehow soluble in water as solute, depending on the temperature, which will bring the organic products dissolved in it. In the other hand, water is soluble in EtOA, which will make the work-up more challenging, especially in industrial scale synthesis. So, do this reactor helped the work-up compared with conventional reactors?

(7) The writing style is confusing, since the authors mentioned the corrosion of chemicals to reactors when they mentioned biomimetic designing idea. Usually, biological systems are using water as intermediates and mild conditions, however, organic reactions use organic solvents, which will be big problems for biomimetic systems. More explanations need to make on the designing of the reactors and organic reactions chosen for testing.

(8) How about the potentials in industrial applications of the newly designed reactor? For the cellular reactor, how to remove the reaction heat resulted during the reactions? And how to control the temperature of the reaction system? And this is very crucial to have the processing safety in industry.

Reviewer #2 (Remarks to the Author):

Cao and Tao presented the use of a cellular reactor to address main challenges of liquid-liquid reactions, such as control of phase interface and reactor corrosion due to the use of strongly

acidic, basic or oxidizing reagents.

The authors dedicated an impressive amount of work to the description of the reactor design and working principles, in particular they carefully analyzed the reactor geometry in order to achieve liquid retention within the cells. They also demonstrated its use in several chemical transformations (alcohol protection with 3,4-2H-dihydropyran; Knoevenagel condensation; thiols dimerization; thioether oxidation). The idea is original and fascinating, however several flaws in the work prevent its publication in Nature Communication. First of all, the absence of meaningful comparisons with known literature approaches makes impossible to assess if these cellular reactors represent an advancement in the field. Specifically, the authors emphasized the beneficial aspects of their cellular reactor, but provided little control experiments to effectively compare its performances with respect to those of a standard liquid-liquid system mixed with a bladed impeller. It is well known that the outcome of liquid-liquid reactions is dependent on all geometrical parameters of the reactor, such as the size, shape and number of blades, the stirring speed, the shape and volume of the vessel and so on. In general, all the system geometrical parameters influence the fluid dynamics within the reactor, thus affecting the reaction outcome. In the present work, all the reactions were stirred at 50 rpm using the same cellular reactor as the impeller, and one control experiment only was performed, using a bladed impeller with the same size and shape of the cellular reactor, again stirring the reaction at 50 rpm. No analysis of fluid dynamics is reported for any of the described cellular reactor geometries. At the present status of this research, no comparison has been reported between the cellular reactor performances and those of any other standard bladed impeller, therefore it is not really possible to comment on the absolute performances.

Moreover, I am not convinced that the authors conclusions are fully supported by the presented data. More specifically:

- regarding the enhancement of phase interface contact by 300 times, no experimental data in the main text of the manuscript is presented supporting this claim. In the Supporting Information, only a paragraph is dedicated to this aspect, where the authors comment the outcome of the reaction between dihydropyran and benzyl alcohol (Table S3, entry 19 and 20). The authors here say that the liquid-liquid contact area for the reaction performed with the cellular reactor is $2,376 \text{ cm}^2$, while the liquid-liquid contact area of the reaction performed with a non-cellular impeller with the same shape and size is only 8.1 cm^2 . No further data is provided in support to those numbers, and, in the case of the cellular reactor, it is not explained if the $2,376 \text{ cm}^2$ surface is measured or calculated on the basis of geometrical considerations. Actually, the liquid-liquid contact area resulting when using the cellular reactor is dependent on the meniscus between the two phases, and the shape of such meniscus will depend on the force applied by rotating the cellular reactor. This means that the liquid-liquid contact area between the phases is likely to be different if the reactor is standing or rotating. Also, cells at different radial distances from the center are likely to have different liquid-liquid contact area during rotation. In view of all this, more details are needed to understand how the liquid-liquid contact area has been evaluated for the cellular reactor.

- regarding the restriction of corrosive components within the reactor, no experimental data are reported at all. The authors in fact convincingly report on the capability of the cellular reactor to hold liquid while standing (Figure S2, and movie 1, 2 and 3), but only movie 4 shows the cellular reactor rotating in hexanes while holding water. Unfortunately, it is not easy to understand from the video if any droplet of aqueous phase separates from the reactor. Moreover, even if no aqueous phase goes in the hexanes (this is likely, as water solubility in hexane is only 0.0036 M), this might not be the case when ethyl acetate is used as the reaction solvent (as it was done for the Knoevenagel condensation, thiols dimerization and thioether oxidation). In fact, ethyl acetate solubility in water (at room temperature) is higher than 8%, and water solubility in ethyl acetate in the same conditions is higher than 3%, therefore I would expect a certain degree of cross-contamination between the two phases (especially as reactions are performed at $60 \text{ }^\circ\text{C}$). However, no experimental data are provided of cross contamination between the two phases, as the products obtained in all the reactions have never been isolated, but only analyzed by GC-MS. Since all the reported transformations afford no reaction byproducts soluble in organic solvent, only the pure product should be obtained after evaporation of the solvent composing the organic phase, thus assessing the degree of cross-contamination should be possible e.g. by TGA in the case of Knoevenagel condensation and thioether oxidation.

At the present status of this work, I would recommend publication on a different journal, after

major revisions. Several aspects should be addressed before considering this work for publication:

- I am not an expert of simulation and modeling, but I do not really think that the paragraph "Modeling of interfacial reactions" is needed for this work. Also, the proposed model is valid for a flat interface between static fluids, which does not correspond to the experimental situation.
- further insight is needed to understand the benefits of using cellular reactors for chemical transformations. For example, comparison of reaction outcomes at varying rotation speeds might offer a starting point to evaluate the performances of the cellular reactor with respect to a standard bladed impeller. Reproducibility of reactions outcome might be a good parameter to evaluate as well (I would expect better reproducibility by using the cellular reactor).
- further insight is also needed to understand the problems/limitations related to the use of cellular reactors for chemical transformations. For example, can this reactor be used to perform reactions leading to products which partition in both phases? Can this reactor be used to perform reactions sensitive to oxygen or carbon dioxide? What problems arise in case of formation of an insoluble product/byproduct (that therefore crushes out from solution)?

As a final remark, the manuscript needs some improvement in terms of clarity, and referencing. At the first read, I had some difficulties in understanding the meaning of reactor parameters L and D as listed in the text, later I found them in figure 2a, but they are not obvious to see (and there is no direct reference in the text to their presence in figure 2a). Maybe changing the color code in order to have all the reactors trusses of the same color in all the four 2a pictures would help. Also, parameters φ (page 7, line 3) and θ (page 8, line 2) are not defined. In figure 2d, parameter G^* is present, but is not defined anywhere in the text. In figure 3, graph c is entirely dedicated to the dependence of parameter λ from D^* , however λ is not clearly defined in the text (it is only presented as a dimensionless number describing the wetting properties of the cell). After description of cellular reactors geometries SC, BCC and BCC with frame, there is no clear indication of which geometry has been used to run the reactions. Finally, when referring to the reusability of the cellular reactor (page 21, line 10), the authors should state in the text what is the reaction they are re-running (it is said in the SI but not in the main text), especially because it is none of the thioether oxidations described in the previous paragraph. Regarding the referencing, very little space is dedicated to the description of the state of the art in literature, the authors should give a broader overview of the field (not only related to Pickering emulsions, e.g. see 10.1021/acscami.8b14321, 10.1021/acsnano.8b05718, 10.1038/s41598-019-42769-8). Also, ref. 6 is placed incorrectly.

Reviewer #3 (Remarks to the Author):

This is an exciting piece of work demonstrating cellular fluidics as a method to perform liquid-liquid reactions. Cellular fluidics has the potential to allow for enhanced interfacial area between the two liquid phases, while maintaining continuous liquid phases (ie no emulsions). In their work, the aqueous phase contains a catalyst or oxidant and is contained within the inert cellular fluidic structure and submerged in an organic phase which contains the substrate. An analysis was performed to predict geometries which would function as intended. A visual demonstration using a pH indicator was used to show the reaction occurring in the cellular fluidic reactor. They demonstrate this new format for liquid-liquid reactions with four different reactions and quantified them with GC-MS.

This work builds on 3D printed structures to order two immiscible fluids for liquid-liquid reactions. While these concepts are not completely novel, this work is significant in demonstrating how these concepts can be used for chemical reactions. Furthermore, the analysis presented suggests a relatively straightforward method to determine if a geometry is stable and can be used for this function.

- More background and comparison to liquid-liquid reactors. I understand if this work does not perform well compared to the state of the art liquid-liquid reactors, but I would

still like to see some comparison and discussion of where this technology can be further improved. How does the time scale for this work compare to the state-of-the-art analytical chemistry tools for liquidliquid extraction where reactions are conducted in glass tubes and that corrosion is not a concern?

The analysis is difficult to follow and missing key definitions and details needed to replicate the work.

- The analysis for eq 1-7 look similar to (22) however, equations 4-7 look different. It is unclear if the adhesive force is normalized? The units do not look correct for eq4 and 6. I am unclear why the surface-liquid area is being integrated over instead of the contact perimeter?
- I believe there is an error in eq9 for the SI $A_s = A_{LE} + A_{SL1} + A_{SL2}$
- I could not follow the analysis after table S1 in the SI. The surface tension data for water and diiomethane is presented in the table, but the analysis is for water and cyclohexane? Its unclear what values are measured and which ones are derived. Plugging in the numbers seems to give different numbers than what is presented. I think it would benefit from a table presenting all the measured values for the different systems used in this work. This may also help clarify the different subscripts. It was incredibly difficult to follow given the three different phases 1, 2, 3 and the liquid/solid subscripts. (i.e. what is liquid 3 after equation 6? In SI)
- **The equivalent wetting factor, λ , suggests an exciting simple method to evaluate different geometries for their ability to hold fluid. Yet, there is little to no description of how this parameter is calculated or measured. Please provide additional information and corresponding experiments (?) necessary to determine the wetting factor.**
- **The molecular dynamics simulations are difficult to follow and weakens this work. This takes away from the paper and opens more questions. For example, the size of the simulation (number of atoms, dimensions of cell) is not specified. The time step units are not specified. It is not clear what the density is normalized by. Fig 4b is misleading- its unclear what this shows. This analysis should be removed.**

More analysis/commentary on:

- A key feature that is barely discussed is the rotation rate. The rotation rate is stated in the methods but there is no discussion about the effect of the rotation, the maximum rate allowable to keep the fluid contained, or even a sentence suggesting a limit to the rotation rate. While I understand if an extensive analysis of the rotation rate is outside the scope, simply calculating the capillary number, and including a discussion of how the rotation rate affects the operation is necessary. What are the reaction rates with no rotation? With rotation?
- Another highlight of this technology is being able to separate and collect the different phases after the reaction. There is very little discussion of how liquid is recovered from the cellular fluidic device. If possible a video of this would be incredibly informative, otherwise a more detailed description is warranted. What is the recovery rate?
- I would really like to see more comparison with experiment to validate Fig 3. By modifying the unit cell size I think it may be possible to print structures above and below the y=x line (ie where the device should and should not work). Plotting 3-5 experimental points on figure 3d would be very helpful. Where is the device you used? A single unit

cell, or a small group of cells would be sufficient for this demonstration and allow you to probe different unit cell types.

- Colorimetric demonstration: Video 1: There looks like there is bubble formation on the left blade for the first half of the video. Fig 4a: Is there an explanation for why the top part the blade turned blue while the bottom stayed clear for the first 60s of the reaction in Fig 4a? Is there displacement of fluid? Other concerns and comments:
- The amount of text (and figure 1) dedicated to drawing parallels between biological systems and this device seems excessive. While I am not opposed to the narrative of being inspired by natural systems, the detailed description of cells, syncytium etc seems unnecessary. Especially since each unit cell is identical in the reactors, it's difficult to think of each of these bionic cells as individual cell types contributing to the whole organism. I was left wanting much more background about chemical liquid-liquid reactors, the current state of the art and the gold standard for performing these types of reactions. I suggest toning down the analogy, and instead going into more detail about liquid-liquid reactors. I would also try to change figure 1 to add more value- perhaps moving something like fig 2a into fig1- or having a general schematic of the different oil/aq phase and inert structure? A schematic of the reactions taking place? The overall concept and experiment was difficult to understand for people not familiar with 3D printed cellular reactors, a visual clearly describing the different phases (organic, aqueous) and the reaction occurring would be very helpful.
- What is the stability of the printed structures? Is etching and aging of the plastic a concern? Any characterization? Why is there a slight increasing trend as the cycle number increases for table S5? Is it perhaps the strut is being etched by the acid and thinning and allowing for more 1M H₂SO₄ to be loaded for the subsequent cycle therefore increasing the conversion rate?
- Bromothymol blue may be redissolving in the cyclohexane (no water transfer just bromothymol blue) and changing back to yellow. Specific concerns
- Pg 9, line 4: "...holding capacity when the characteristic size L surpasses h." L is defined as the unit cell size—does this indicate that when the unit cell size surpasses h the cell cannot hold liquid regardless of the unit cell type? I would expect unit cell types with high relative density (large amount of internal struts) to be able to hold fluid even if the overall unit cell size is greater than h. I think the thought behind this sentence is correct, but specifying L is confusing.
- Specify which reactors are used in each reaction
- Pg 16 line 18: "metal reactor": isn't the reactor plastic?
- How does the contact angle change with temperature given the elevated temperature of some of your reactions
- Fig 3d: typo cubic, n =3
- Fig 3b and 3f: keep color scheme consistent
- I do not see any captions or descriptions for videos! Overall, I think the operation of this device is very exciting, but the videos don't capture the beauty of this application. Was the colorimetric reaction done statically or with rotation?

Please consider reviewing and citing : <https://doi.org/10.1038/s41567-021-01204-4>.

Reviewer #4 (Remarks to the Author):

This paper presents a biomimetic 3D printed cell reactor for liquid-liquid reactions, offering intriguing results that, with some minor revisions, I believe can be suitable for publication. Below

are specific comments and suggestions for improvement:

1. On page 15, line 8, the authors posit that the product is primarily soluble in the oil phase. However, in Figure 4a, a water-soluble product is shown transferring to water. It would be beneficial if an experiment illustrating the transfer of the product to the oil phase could be included.
2. In the Introduction section, it is advisable for the authors to include references to application-specific literature to clarify the engineering significance of their results.
3. Could the authors provide a description of the separation process that follows the reaction and elaborate on the typical methods used for product separation?
4. The authors are advised to add specific findings to the Conclusion section.
5. Have the MD simulations been conducted using the VASP software? For the MD simulations, it is recommended that the authors include details about the algorithms used for temperature and pressure control, information about how the system reached equilibrium, and the time step employed.

Reviewer #5 (Remarks to the Author):

This was a joint review conducted with one of the other reviewers; all comments are already contained in the other review.

Reviewer #6 (Remarks to the Author):

This was a joint review conducted with one of the other reviewers; all comments are already contained in the other review.

RESPONSE TO REVIEWERS' COMMENTS

Comments from the Reviewer:

Reviewer: 1

Overall Comments:

In this paper, Jinzhe Cao et. al. reported an a biomimetic 3D-printed cellular reactor for organic reactions such as acetal formation reaction between 3,4 - 2H-dihydropyran and primary alcohols under acidic conditions, Knoevenagel condensation reaction under basic conditions, oxidation reactions of thiols by hydroperoxide and oxidation reactions on thiol ethers by HIO₄. The authors described the design and manufacture of the cellular reactors by 3D-printing, however, however, it is very hard to judge whether the new reactors possessed some enhancements for selected organic transformations compared with conventional reactors. It is suggested major revision and resent for review. Some other points need to be addressed as following:

Response: We appreciate the reviewer's highly positive comments. We tried our best to address all the detailed comments and made a detailed revision of our manuscript.

Q1 The cellular reactors in this paper are made of resin FLGPCL04 and some treatments need done to make them hydrophilic and hydrophobic. Is the resin stable enough when it is contacted with hot organic solvents under acidic, basic and oxidative conditions? It is worth investigating if the cellular reactors remained the same after several cycles of reactions. And any other materials than resin FLGPCL04 are suitable for 3D printing and the selected reactions?

Response: Thanks the reviewer for the valuable comments. FLGPCL04 is essentially a photopolymer resin, serving as a cost-effective and versatile

commercial resin material suitable for various applications. Utilizing FLGPCL04 resin in research allows for a more economical investigation of the fundamental guidelines for liquid retention by the cellular reactor and the structure-activity relationship of the cellular reactor. The cellular reactor demonstrates strong tolerance when in contact with strong acids, strong bases, strong oxidants, and hot organic solvents. Photographs and SEM images indicate that after five cycles of stability experiments, the cellular reactor maintained its complete appearance and stable morphology.

To further investigate the stability of different resins in solvent systems, during the supplementary experiments, we purchased two additional commercial resins from Formlabs. Information on the three resins can be found in the table below.

Table S9 Introduction to the three resins

Name	Part Number	Description
Clear Resin v4	FLGPCL04	Clear resin, general purpose
Tough 2000 Resin v1	RS-F2-TO20-01	High strength resin
High Temp Resin v2	RS-F2-HTAM-02	High temperature resistant resin

To further investigate the corrosive effects of various components on cellular reactor materials, flat cubic samples made from the aforementioned three resins were 3D printed, with dimensions of 10 mm x 10 mm x 2 mm. The flat structure facilitates SEM testing and allows a more accurate comparison of surface morphology changes before and after treatment.

Figure S8 Corrosion resistance testing of printed samples from three resins: (a) Small pieces of the three resins were placed in sample bottles containing 50 mL of corrosive liquid and heated at 60°C for 12 hours. (b) Appearance of the small pieces in their original state, untreated. (c) Appearance of the small pieces after heating in a glass surface dish. (d-j) Appearance of the small pieces after treatment with various corrosive liquids.

The following test was conducted to observe the potential corrosion of cellular reactor materials by corrosive components during the catalytic reaction process. Each sample bottle (100 mL) contained three small pieces printed from different resins and 50 mL of corrosive liquid, with parallel tests conducted using seven

different corrosive liquids. Additionally, to assess the impact of heating on the cellular reactor, three small pieces were placed in a glass surface dish and heated alongside the aforementioned seven sample bottles in an oven at 60°C for 12 hours. The seven corrosive liquids tested were 1M H₂SO₄, 1M NaOH, 30 wt% H₂O₂, 1M H₅IO₆, ethyl acetate, n-hexane, and cyclohexane.

Initially, from a visual perspective, whether heated directly or in corrosive liquids, there were no significant changes in the appearance of the small pieces before and after treatment (Fig S8 b-j).

Figure S9 SEM images of clear resin printed sheets (a) (a) raw state, without any processing (b) heated in glass surface dish (c) processed by 1M H₂SO₄ (d) processed by 1M NaOH (e) processed by 30 wt% H₂O₂ (f) processed by 1M H₅IO₆ (g) processed by ethyl acetate (h) processed by n-hexane (i) processed

by cyclohexane.

Initially, from a visual perspective, whether heated directly or in corrosive liquids, there were no significant changes in the appearance of the small pieces before and after treatment (Fig S8b-j).

Figure S10 SEM images of tough-2000 resin printed sheets (a) raw state, without any processing (b) heated in glass surface dish (c) processed by 1M H_2SO_4 (d) processed by 1M NaOH (e) processed by 30 wt% H_2O_2 (f) processed by 1M H_5IO_6 (g) processed by ethyl acetate (h) processed by n-hexane (i) processed by cyclohexane.

From the SEM images, the small pieces before treatment (Fig. S9a) exhibited

a grooved structure on the surface due to the layer-by-layer manufacturing process in additive manufacturing. After treatment, the small clear resin pieces (Fig. S9b-i), compared to the untreated ones, maintained their grooved structure in the vast majority, indicating that the aforementioned treatments had minimal corrosive effects on the clear resin. The only exception was the small piece treated with NaOH solution, which showed a more flattened grooved structure, suggesting that clear resin has a lower tolerance to strong alkalis.

The situation of the high-strength resin pieces was similar to that of the clear resin pieces (Fig. S10a-i), with strong alkalis causing some degree of corrosion to the high-strength resin.

Figure S11 SEM images of high-temp resin printed sheets (a) raw state, without

any processing (b) heated in glass surface dish (c) processed by 1M H₂SO₄ (d) processed by 1M NaOH (e) processed by 30 wt% H₂O₂ (f) processed by 1M H₅IO₆ (g) processed by ethyl acetate (h) processed by n-hexane (i) processed by cyclohexane.

Surprisingly, for the high-temperature resistant resin, the surfaces of the treated small pieces (Fig. S11b-i) showed almost no changes compared to the untreated ones (Fig. S11a). Only the pieces treated with n-hexane and cyclohexane (Fig. S11h-i) slightly collapsed in the surface grooves, but the microstructure (shown in the small image in the top left corner) remained unchanged. These observations indicate that the high-temperature resistant resin can withstand corrosion from all the aforementioned solutions, making it a widely applicable resin material.

To better describe the cellular reactor material's tolerance to NaOH, we employed static tensile testing to characterize the material's elastic modulus change before and after treatment with corrosive components. Static tensile testing is one of the most commonly recognized methods for measuring elastic modulus. The test involves placing the specimen in a tensile testing machine and measuring the material's strain and stress under a specified tensile force. Strain is defined as the ratio of deformation length in the direction of tension to the total length, a dimensionless number, and the force corresponding to this

strain is defined as stress.

Figure S12 presents the testing of resin materials for NaOH tolerance: (a) the blueprint of the standard test specimen, (b) a photograph of the standard test specimen, and (c) the strain-stress curves of the resin materials before and after NaOH treatment.

Prefix CR represents Clear Resin v4, and TR stands for Tough 2000 Resin v1. The suffix raw indicates the original, untreated standard test specimens, while NaOH denotes the specimens after being heated in 1M NaOH at 60°C for 12 hours. In the strain-stress curves, stress increases with strain during the initial stretching phase, characterizing the elastic deformation process. When strain reaches a certain critical value, stress decreases with increasing strain, marking the plastic deformation stage, where the deformation becomes irreversible. Upon reaching another critical strain value, the stress drops precipitously, indicating the specimen's breakage. For the standard specimens of Clear Resin, both elastic and plastic deformation capabilities decreased after NaOH

treatment, yet they retained considerable elastic deformation capacity. Similarly, the standard specimens of Tough Resin yielded consistent conclusions.

Therefore, whether it is clear resin, high-strength resin, or high-temperature resistant resin, all exhibit good corrosion resistance. Unless the reaction is extremely sensitive and requires high-temperature resistant resin, the general and cost-effective clear resin remains the best choice.

The relevant content has been added to the Supporting Information.

Q2 In this paper, GC-MS was used to monitor the reaction process. However, that is not enough. HPLC and NMR are highly suggested, in addition to GC-MS, especially for some compounds which are not volatile.

Response: Thanks the reviewer for the valuable comments. The compounds synthesized in this study are commonly reported in the literature and commercially available. Hence, GC-MS can provide satisfactory detection results. Moreover, GC-MS testing is more convenient and accurate for aliphatic substrates without chromophores than HPLC. This testing method is also widely reported in the literature (<https://doi.org/10.1021/jacs.6b04265> , [10.1038/s41929-020-00526-5](https://doi.org/10.1038/s41929-020-00526-5) , [10.1038/ncomms9580](https://doi.org/10.1038/ncomms9580) , <https://doi.org/10.1002/anie.201803464>).

Of course, for reactions involving substrates/products with chromophores, HPLC detection is entirely suitable, and we have supplemented this with UPLC-MS testing to prove the successful synthesis of the aforementioned compounds. Similarly, we conducted HMR testing on some of the reaction products, with relevant data referred to in Fig S39. NMR test results indicate the successful synthesis of the aforementioned compounds.

Figure S39 UPLC-MS spectrum of entry 17

Figure S20 NMR spectrum of Benzylidenemalononitrle

^1H NMR (400 MHz, CDCl_3) δ 7.91 (d, $J = 7.4$ Hz, 2H), 7.78 (s, 1H), 7.64 (t, $J = 7.4$ Hz, 1H), 7.55 (t, $J = 7.6$ Hz, 2H).

Figure S25 NMR spectrum of Di-n-hexyl-disulfide

^1H NMR (400 MHz, CDCl_3) δ 2.73 – 2.62 (m, 5H), 1.72 – 1.62 (m, 3H), 1.44 – 1.23 (m, 13H), 0.93 – 0.83 (m, 5H).

Figure S28 NMR spectrum of 4,4'-dimethyldiphenyldisulphide

¹H NMR (400 MHz, CDCl₃) δ 7.38 (d, J = 8.3 Hz, 3H), 7.10 (d, J = 8.1 Hz, 4H), 2.32 (s, 7H).

Figure S30 NMR spectrum of Di-4-fluorophenyl sulfide

¹H NMR (400 MHz, CDCl₃) δ 7.36 (dd, J = 8.9, 5.1 Hz, 4H), 6.92 (t, J = 8.7 Hz, 4H).

Figure S32 NMR spectrum of Tetramethylene sulfoxide

^1H NMR (400 MHz, CDCl_3) δ 2.97 – 2.77 (m, 4H), 2.55 – 2.35 (m, 4H), 2.11 – 1.96 (m, 4H).

Figure S34 NMR spectrum of Methyl phenyl sulfone

^1H NMR (400 MHz, CDCl_3) δ 7.96 (d, $J = 7.0$ Hz, 1H), 7.67 (t, $J = 7.4$ Hz, 1H), 7.58 (t, $J = 7.4$ Hz, 2H), 3.06 (s, 4H).

Figure S36 NMR spectrum of Phenyl vinyl sulfone

¹H NMR (400 MHz, CDCl₃) δ 7.90 (d, J = 7.0 Hz, 1H), 7.64 (t, J = 7.4 Hz, 1H), 7.56 (t, J = 7.4 Hz, 2H), 6.72 – 6.61 (m, 1H), 6.47 (d, J = 16.6 Hz, 1H), 6.04 (d, J = 9.8 Hz, 1H).

Figure S38 NMR spectrum of Di-n-butyl-disulfide

¹H NMR (400 MHz, CDCl₃) δ 3.02 – 2.88 (m, 4H), 1.89 – 1.73 (m, 4H), 1.48 (h, J = 7.4 Hz, 4H), 0.97 (t, J = 7.3 Hz, 6H).

Q3 It is very hard to judge whether the new reactors possessed some enhancements for selected organic transformations compared with conventional reactors. Some metrics need to be proposed to evaluate this point, other than conversions and yields of the well-developed organic reactions.

Response: Thanks the reviewer for the valuable comments. The reactor we designed aims to facilitate water-oil biphasic reactions involving corrosive substances in water, simplifying the post-reaction processing challenges and easing the separation of water and oil phases thereafter. Hence, we selected four categories of acids, bases, hydrogen peroxide, and strong oxidants. These

are commonly used chemical reagents in organic synthesis, especially in synthesizing fine chemicals, representing the types of chemicals typically encountered.

Q4 For the oxidation of thioethers, not only to see the conversions of substrates, the confirmation of the products is very crucial. Sulfones and sulfoxides are both possibly obtained? Please give more detailed information.

Response: Thanks the reviewer for the valuable comments. This study used periodic acid as the oxidizing agent to oxidize thioether substrates, producing sulfoxides or sulfones as products. The difference of one oxygen atom between the corresponding sulfoxide and sulfone molecules of the same thioether results in variations in their physicochemical properties, including molecular weight, boiling point, and polarity. Under the same GC-MS chromatographic conditions, there are significant differences in their retention times. Additionally, we conducted NMR testing on some of the products, confirming that the products are indeed as described in the main text. Entries 13-16 products are sulfoxide, sulfone, sulfone, and sulfone. NMR spectra are provided below.

Figure S32 NMR spectrum of Tetramethylene sulfoxide

^1H NMR (400 MHz, CDCl_3) δ 2.97 – 2.77 (m, 4H), 2.55 – 2.35 (m, 4H), 2.11 – 1.96 (m, 4H).

Figure S34 NMR spectrum of Methyl phenyl sulfone

^1H NMR (400 MHz, CDCl_3) δ 7.96 (d, $J = 7.0$ Hz, 1H), 7.67 (t, $J = 7.4$ Hz, 1H), 7.58 (t, $J = 7.4$ Hz, 2H), 3.06 (s, 4H).

Figure S36 NMR spectrum of Phenyl vinyl sulfone

¹H NMR (400 MHz, CDCl₃) δ 7.90 (d, J = 7.0 Hz, 1H), 7.64 (t, J = 7.4 Hz, 1H), 7.56 (t, J = 7.4 Hz, 2H), 6.72 – 6.61 (m, 1H), 6.47 (d, J = 16.6 Hz, 1H), 6.04 (d, J = 9.8 Hz, 1H).

Figure S38 NMR spectrum of Di-n-butyl-disulfide

^1H NMR (400 MHz, CDCl_3) δ 3.02 – 2.88 (m, 4H), 1.89 – 1.73 (m, 4H), 1.48 (h, $J = 7.4$ Hz, 4H), 0.97 (t, $J = 7.3$ Hz, 6H).

Q5 The title actually can not show what has been done in the paper. Some more connections between the newly constructed reactors and the well-studied organic reactions are suggested to be strengthened. At this moment, the two parts of work are just put together, as many more reactions could be taken and tested, but why just the four reactions were tested by this new reactor.

Response: Thanks the reviewer for the valuable suggestions. In this paper, one of the key roles of the cellular reactor is to restrict the flow of the internal phase liquid and to maintain the stability of the phase interface. This provides a venue for interfacial chemical reactions to occur while ensuring that the corrosive

components of the internal phase do not come into contact with or adhere to the reactor walls. Under this premise, the distribution of substrates and catalyst components across the two phases is an indispensable condition, thereby excluding certain organic reactions. Moreover, many organic reactions require anhydrous conditions, and an aqueous phase significantly reduces the range of selectable reactions. Additionally, traditional water-oil biphasic reactions often require vigorous stirring to form an emulsion system to enhance mass transfer. However, when the aqueous phase contains substances like acids, bases, or strong oxidants, the subsequent separation process becomes more complicated and poses significant safety risks. For instance, neutralizing acids, bases, or quenching oxidants may be necessary before further processing. In this work, lifting the cellular reactor out of the organic solution after the reaction can complete the post-reaction separation, greatly simplifying this process. Therefore, we selected reactions involving these types of corrosive substances. Also, to enhance the readers' understanding of our article, we have revised the title. The revised title is as follows. "*Liquid-Liquid Reactions Performed by Cellular Reactors*"

Q6 The liquid-liquid reactions are complex beside the simple phase separations in fact. Take the water and EtOAc as example, in aqueous phase, EtOAc was somehow soluble in water as solute, depending on the temperature, which will bring the organic products dissolved in it. In the other hand, water is soluble in EtOAc, which will make the work-up more challenging, especially in industrial scale synthesis. So, do this reactor helped the work-up compared with conventional reactors?

Response: Thanks the reviewer for the valuable comments, which is crucial for the improvement of our work. After the reaction concludes, the inward liquid being confined within the cellular reactor allows for the separation of the two phases by simply lifting the cellular reactor out of the reaction vessel. Operationally, the cellular reactor indeed reduces the steps involved in phase

separation, contributing to savings in subsequent processing stages.

Regarding the issue of cross-contamination between the two phases you mentioned, in biphasic reactions involving water and low-polarity solvents (such as hexane and cyclohexane), the disparity in solubility between water and these solvents is indeed significant. The solubility of hexane in water is 0.014%, making cross-contamination between the phases negligible. We measured the relative content between the two phases, such as water and ethyl acetate (EtOAc), for systems where the two phases are slightly soluble. The content of EtOAc in water was determined by Total Organic Carbon (TOC) analysis, while the water content in EtOAc was measured using NMR.

The cellular reactor first adsorbs water and is then inserted into 50 mL of ethyl acetate, after stirring at 50 rpm at room temperature. For 5 hours, the water phase is separated from the cellular reactor, yielding sample 1W, with the organic phase being sample 1EA. Using a commercially available stirrer, an amount of water equal to the holding capacity of the cellular reactor is added to 50 mL of ethyl acetate. After stirring at 50 rpm at room temperature for 5 hours, the water phase is extracted to obtain sample 2W, with the organic phase being sample 2EA. The process for obtaining sample 3 is identical to that of sample 2, except the stirring speed is changed from 50 rpm to 1000 rpm. After stirring and allowing the phases to separate, the water phase obtained is sample 3W, with the organic phase sample 3EA. The cellular reactor is made of resin material and could also be a potential organic carbon source. Therefore, we placed the cellular reactor in 50mL of water and stirred at 50 rpm at 60°C for 5 hours to verify whether the cellular reactor would contaminate the water phase, with the sample being labeled as 4W.

For the content of ethyl acetate in water, we approximate its concentration by measuring the Total Organic Carbon content using a TOC analyzer. The TOC analyzer employs a combustion method, oxidizing all organic substances to carbon dioxide, determining the organic carbon content.

Table S15 Total organic carbon in water

Sample	DI Water	S15-1W	S15-2W	S15-3W	S15-4W
TOC/ppm	2.13	1467	2425.5	1366	4.57

As a blank control, the TOC content in the pure deionized water used throughout the experiment was 2.13 ppm. Sample 4W showed a TOC content of 4.57 ppm for a second blank control, indicating that the cellular reactor released a minimal amount of organic substances into the water. This further demonstrates the stability of the cellular reactor in water. Under ambient temperature and stirring at 50 rpm, sample 1, obtained using the cellular reactor, had a lower ethyl acetate content in water compared to sample 2, obtained using a commercial stirrer, signifying that the use of the cellular reactor can significantly reduce cross-contamination between the two phases.

Nuclear magnetic resonance (NMR) spectroscopy provides the water content in ethyl acetate. The fundamental principle of quantitative NMR is as follows: under the same test conditions, the peak area is directly proportional to the total number of hydrogen atoms corresponding to that peak. Therefore, the peak area is also proportional to the amount of substance and the equivalent number of hydrogen atoms. The calculation formula is as follows:

$$C_W = \frac{I_W/n_{WH}}{I_W/n_{WH} + I_{EA}/n_{EAH}}$$

In this context, C_W represents the mole fraction of water in ethyl acetate, I_W is the area of the peak corresponding to water, I_{EA} is the area of the peak corresponding to ethyl acetate, n_{WH} is the number of equivalent hydrogen atoms for the water peak, and n_{EAH} is the number of equivalent hydrogen atoms for the ethyl acetate peak. When quantifying using the hydrogen atoms on the methylene group of ethyl acetate, the formula is adjusted accordingly:

$$C_W = \frac{I_W/2}{I_W/2 + I_{EA}/2} = \frac{I_W}{I_W + I_{EA}}$$

The NMR data for samples S15-1EA to 3EA are as follows:

Figure S40 NMR of sample S15-1EA

Figure S41 NMR of sample S15-2EA

Figure S42 NMR of sample S15-3EA

Table S16 NMR of samples with water content in ethyl acetate

Sample	S15-1EA	S15-2 EA	S15-3 EA
I_W	656.87	607.38	923.58
I_{EA}	6800.13	6750.85	6809.8
C_W	8.8%	8.3%	12.1%

Under ambient conditions, the water content in ethyl acetate using the cellular reactor (S1 EA) is comparable to that without using the cellular reactor (S2 EA) but significantly lower than under vigorous stirring conditions (S3 EA). This indicates that while the cellular reactor increases the interfacial area, it does not introduce additional cross-contamination between the two phases. On the contrary, compared to vigorous stirring, the use of the cellular reactor actually reduces cross-contamination between the phases and facilitates rapid separation afterwards, making it an advanced tool for chemical processes.

In the testing process for water content in ethyl acetate, samples were not

diluted with deuterated reagents; instead, ethyl acetate was directly analyzed. Given that the water content in ethyl acetate is already minimal, dilution with deuterated reagents would increase the relative error in water content measurements due to the presence of small amounts of water in the deuterated reagents themselves. Therefore, deuterated reagents were not introduced in this part of the testing.

Q7 The writing style is confusing, since the authors mentioned the corrosion of chemicals to reactors when they mentioned biomimetic designing idea. Usually, biological systems are using water as intermediates and mild conditions, however, organic reactions use organic solvents, which will be big problems for biomimetic systems. More explanations need to make on the designing of the reactors and organic reactions chosen for testing.

Response: Thanks the reviewer for the valuable comments. We apologize for not clearly articulating the significance of this aspect in our manuscript, which has now been revised. Our work aims to emulate the role of membrane systems in biological organisms during reactions. Membrane systems are widespread within organisms, covering a vast area. Molecules with poor water solubility always distribute themselves on biological membranes. For example, chlorophyll, chemically a porphyrin derivative, is distributed on the thylakoid membranes within chloroplasts. This distribution increases the specific surface area and lipid components of the biological membrane system, significantly enhancing chlorophyll's solubility. Many biological reactions occur through the interface of membranes. Similarly, the main reaction site in our cellular reactor is the transitional layer at the interface between the water and oil phases.

Unlike biochemical reactions in organisms that occur at trace concentrations, the concentration of catalytic reactions in the laboratory is too high, and the solubility provided by the water-oil interface alone is insufficient. Hence, using an organic solvent as the external phase is necessary, primarily to increase the solubility of substrate molecules. Chemical reactions still occur at the water-oil

interface, with substrate molecules in the organic phase not participating in the reaction at that moment but serving as a substrate reserve for subsequent reactions. Moreover, the organic solvent provides a substrate reserve and facilitates product diffusion from the water-oil interface to the organic phase.

In summary, we still believe that the design of our reactor shares many logical similarities with biochemical reactions on biological membranes. We have supplemented this part in the main text accordingly. Please refer to the Background section in the manuscript. The content is as follows.

Liquid-liquid reactions are vital in chemical synthesis due to the fast mass transfer between interfaces, uniform temperature field distribution, dynamic interfacial composition updating, and resistance to catalyst poisoning. Typically, one reactant acts as a dispersed phase, forming tiny droplets in the continuous phase of another substance, increasing the area for molecular diffusion. The surface tension causes the dispersed-phase droplets to adopt a spherical shape, minimizing the system's energy. However, collisions between droplets lead to reduced contact between the liquid phases, often necessitating fast mechanical stirring or emulsifiers.

Stirred reactors are widely used in chemical engineering and industrial processes for liquid-liquid reactions. Stirring facilitates thorough mixing of the two phases, enhancing reaction rates. It also helps maintain temperature uniformity, preventing temperature gradients due to exothermic or endothermic reactions. However, stirred reactors have drawbacks, such as high shear forces that can damage shear-sensitive chemicals and insufficient stirring speeds that fail to maintain adequate interfacial area between the two liquid phases, leading to reduced reaction rates.

Using emulsifiers stabilizes droplets and reduces their size, thereby increasing the total interfacial area. However, the compatibility of emulsifiers with the reaction system must be considered, as the introduction of inappropriate emulsifiers can decrease reaction efficiency. Additionally, the resulting

emulsion complicates the separation and purification of products post-reaction. Most importantly, the use of emulsifiers inevitably requires rapid mechanical stirring.

Phase transfer catalysts can enhance the efficiency of liquid-liquid biphasic reactions without requiring vigorous stirring. However, the subsequent separation process becomes more complex due to the high affinity of phase transfer catalysts for both phases. Therefore, designing a novel reactor that can enhance liquid-liquid interfacial mass transfer at low stirring speeds and allow for easy separation after the reaction is crucial.

In nature, numerous liquid-liquid reactions exhibit remarkable controllability. Cell membranes represent a widely occurring natural interface for such reactions, where reactants diffuse to the interface, and products depart, maintaining interface stability throughout the process. Syncytia, large cellular structures formed by the fusion of multiple cells, facilitate intercellular cooperation and functional integration, enhancing cellular stability and efficiency. Inspired by the structure of syncytia and combining this with the structure of stirred reactors, we propose a novel type of cellular reactor. This design merges multiple units into a single entity, enhancing strength and centralizing functions.

Q8 How about the potentials in industrial applications of the newly designed reactor? For the cellular reactor, how to remove the reaction heat resulted during the reactions? And how to control the temperature of the reaction system? And this is very crucial to have the processing safety in industry.

Response: The cellular reactor is placed inside the reaction vessel, and since most reactions require heating, the temperature of the reaction vessel is controlled through an oil bath. For reactions at room temperature, the reaction rate between interfaces is generally slow, and the amount of reaction heat released per unit of time is limited. During stirring, most of the reaction heat is immediately dispersed into the external phase liquid, with a small portion

possibly absorbed by the internal phase liquid and the reactor. In practice, we also used a condenser to condense and reflux the volatile hot organic vapors, and the condensation and reflux process may likewise help prevent the system from overheating. The thermal transfer system of the cellular reactor can fully utilize the existing heat transfer systems and technological methods of batch reactions. Therefore, it does not pose additional safety risks.

Reviewer: 2

Overall Comments:

Cao and Tao presented the use of a cellular reactor to address main challenges of liquid-liquid reactions, such as control of phase interface and reactor corrosion due to the use of strongly acidic, basic or oxidizing reagents.

The authors dedicated an impressive amount of work to the description of the reactor design and working principles, in particular they carefully analyzed the reactor geometry in order to achieve liquid retention within the cells. They also demonstrated its use in several chemical transformations (alcohol protection with 3,4-dihydropyran; Knoevenagel condensation; thiols dimerization; thioether oxidation). The idea is original and fascinating, however several flaws in the work prevent its publication in Nature Communication.

Response: We appreciate the reviewer's highly positive comments. We tried our best to address all the detailed comments and made a detailed revision of our manuscript.

Q1 First of all, the absence of meaningful comparisons with known literature approaches makes impossible to assess if these cellular reactors represent an advancement in the field. Specifically, the authors emphasized the beneficial aspects of their cellular reactor, but provided little control experiments to effectively compare its performances with respect to those of a standard liquid-liquid system mixed with a bladed impeller. It is well known that the outcome of liquid-liquid reactions is dependent on all geometrical parameters of the reactor,

such as the size, shape and number of blades, the stirring speed, the shape and volume of the vessel and so on. In general, all the system geometrical parameters influence the fluid dynamics within the reactor, thus affecting the reaction outcome. In the present work, all the reactions were stirred at 50 rpm using the same cellular reactor as the impeller, and one control experiment only was performed, using a bladed impeller with the same size and shape of the cellular reactor, again stirring the reaction at 50 rpm. No analysis of fluid dynamics is reported for any of the described cellular reactor geometries. At the present status of this research, no comparison has been reported between the cellular reactor performances and those of any other standard bladed impeller, therefore it is not really possible to comment on the absolute performances.

Response: Thanks the reviewer for the valuable suggestions. We have added particle image velocimetry experiments to compare the fluid dynamics properties of the cellular reactor with those of commercially available generic stirring paddles.

Figure S6 Schematic of Particle Image Velocimetry (a) Diagram of the setup (b) Frontal view with dimension parameters annotated. The particle image velocimetry setup consists of a laser, a high-speed camera, and a synchronizer, including a stirring device related to the cellular reactor. The involved dimensions are as follows:

Table S5 Annotation of dimensions and comments.

Symbol	Length	Description
l_{P1}	12 cm	Length of the container
l_{P2}	12 cm	Width of the container
l_{P3}	7 cm	Height of liquid
l_{P4}	4 cm	Distance between the lower edge of the cellular reactor and the bottom of the liquid
l_{P5}	1 cm	Distance between the lower edge of the cellular reactor and the incident laser light

Figure S7 Particle Image Velocimetry data for the cellular I reactor (Reactor-C) at 50 rpm includes: (a) velocity map and (b) vorticity map. For the commercially available generic stirring paddle 1 (Reactor-G) at 50 rpm: (c) velocity map and

(d) vorticity map. For the commercially available generic stirring paddle 2 (Reactor-H) at 50 rpm: (e) velocity map and (f) vorticity map. For the cellular reactor (Reactor-C) at 100 rpm: (g) velocity map and (h) vorticity map.

The classical PIV (Particle Image Velocimetry) test procedure is as follows: First, assemble the apparatus, ensuring the relative positions of each device are fixed, using a level to maintain the laser's perpendicular orientation to the incident plane and the high-speed camera. Fluorescently labeled tracer particles are added to hexane and sonicated to ensure uniform dispersion. The aforementioned mixture is then added to a container, followed by water adsorption into the cellular reactor, which is then inserted into the corresponding position in the container and rotated at an appropriate angular velocity. After stabilizing the rotation for a period, the laser and camera are activated to collect images. The collected images are analyzed using the free, open-source PIV Lab code in Matlab software to determine fluid motion based on the flow of the particles, obtaining images and data on velocity and vorticity. The procedure for using a generic stirring paddle is identical to the above, except that the cellular reactor is replaced with a generic stirring paddle, and water is not adsorbed.

Table S6 Maximum velocity and maximum vorticity as measured by PIV

Reactor	No.	Rotation speed	Maximum velocity (m/s)	Maximum vorticity (1/s)
Reactor-C	a&b	50 rpm	0.04	32
Reactor-G	c&d	50 rpm	0.03	15
Reactor-H	e&f	50 rpm	0.036	16
Reactor-C	g&h	100 rpm	0.09	80

Based on the data in Table S6, at the same rotational speed, the cellular reactor can induce greater velocity and larger flow field disturbances in the reaction system, outperforming the fluid dynamics capabilities of two types of generic stirring paddles. Furthermore, the cellular reactor possesses liquid retention and separation functions that generic stirring paddles lack. Therefore, the design of the cellular reactor is meaningful.

Q2 Moreover, I am not convinced that the authors conclusions are fully supported by the presented data. More specifically:

- regarding the enhancement of phase interface contact by 300 times, no experimental data in the main text of the manuscript is presented supporting this claim. In the Supporting Information, only a paragraph is dedicated to this aspect, where the authors comment the outcome of the reaction between dihydropyran and benzyl alcohol (Table S3, entry 19 and 20). The authors here say that the liquid-liquid contact area for the reaction performed with the cellular reactor is 2,376 cm², while the liquid-liquid contact area of the reaction performed with a non-cellular impeller with the same shape and size is only 8.1 cm². No further data is provided in support to those numbers, and, in the case of the cellular reactor, it is not explained if the 2,376 cm² surface is measured or calculated on the basis of geometrical considerations. Actually, the liquid-liquid contact area resulting when using the cellular reactor is dependent on the meniscus between the two phases, and the shape of such meniscus will depend on the force applied by rotating the cellular reactor.

This means that the liquid-liquid contact area between the phases is likely to be different if the reactor is standing or rotating. Also, cells at different radial distances from the center are likely to have different liquid-liquid contact area during rotation. In view of all this, more details are needed to understand how the liquid-liquid contact area has been evaluated for the cellular reactor.

Response: Thanks for the valuable suggestion from the reviewer. We apologize for not clearly elucidating the aforementioned issue in the article, and now

provide the following explanation:

The data regarding the interphase interface is derived from calculations. Initially, the system utilized a solid stirring paddle at a low speed (50 rpm). To compare the contact area and characterize the benefits brought by using the cellular reactor, the volume of the aqueous phase catalytic liquid added was consistent with the liquid holding capacity of the cellular reactor. Within the reaction liquid, the aqueous phase components assume a shape very close to spherical contraction. Hence, we employed the formula for calculating the surface area of a sphere to estimate the contact area between the two phases..

$$S = 4\pi R^2$$

$$V = \frac{4}{3}\pi R^3$$

Herein, S represents the surface area of the spherical droplet, R denotes the radius of the spherical droplet, and V signifies the volume of the spherical droplet. This gives an interfacial area of 0.00081 mm².

We have re-examined the calculation process and corrected the calculations as follows:

Furthermore, regarding the liquid-liquid catalytic process involving the cellular reactor. In practice, the internal phase liquid is adsorbed into and fills the interior of the cellular reactor. Thus, the interfacial area is determined by the geometric shape of the cellular reactor.

$$S_{LL} = \frac{N \times A_{LL}^*}{6}$$

Here, S_{LL} represents the liquid-liquid interface area, N denotes the total number of degrees of freedom for the cellular reactor, and 6 represents the number of faces on a cubic cell (since A_{LL}^* is determined by the sum of all faces within the cell). The total area calculated from this is 0.00184 m².

The relevant content has been added to the supporting information.

Calculation of liquid-liquid interfacial area created by cellular reactors with and

without cellular units.

Table S12 Catalytic performance of different fractionated cellular reactors

Entry	Recator	Time 1	Conversion rate 1	Time 2	Conversion rate 2
20	S4-C	60 min	50.8%	120 min	83.6%
21	S4-F	60 min	2.48%	90 min	4.86%

The use of the cellular reactor increased the interfacial area between the two phases by 2.3 times and enhanced the reaction rate. Compared to using a solid stirring paddle, the use of the cellular reactor significantly improved the conversion rate within the same period.

Q3 regarding the restriction of corrosive components within the reactor, no experimental data are reported at all. The authors in fact convincingly report on the capability of the cellular reactor to hold liquid while standing (Figure S2, and movie 1, 2 and 3), but only movie 4 shows the cellular reactor rotating in hexanes while holding water. Unfortunately, it is not easy to understand from the video if any droplet of aqueous phase separates from the reactor. Moreover, even if no aqueous phase goes in the hexanes (this is likely, as water solubility in hexane is only 0.0036 M), this might not be the case when ethyl acetate is used as the reaction solvent (as it was done for the Knoevenagel condensation, thiols dimerization and thioether oxidation). In fact, ethyl acetate solubility in water (at room temperature) is higher than 8%, and water solubility in ethyl acetate in the same conditions is higher than 3%, therefore I would expect a certain degree of cross-contamination between the two phases (especially as reactions are performed at 60°C). However, no experimental data are provided of cross contamination between the two phases, as the products obtained in all

the reactions have never been isolated, but only analyzed by GC-MS. Since all the reported transformations afford no reaction byproducts soluble in organic solvent, only the pure product should be obtained after evaporation of the solvent composing the organic phase, thus assessing the degree of cross-contamination should be possible e.g. by TGA in the case of Knoevenagel condensation and thioether oxidation.

Response: Thanks the reviewer for the valuable suggestions. In response to your advice, we have supplemented our work with additional experimental activities.

The capability of the cellular reactor to retain liquid under static conditions has been confirmed. We have also provided additional videos to demonstrate the liquid-holding capacity of the cellular reactor under rotational conditions, as referenced in video Movie 5.

After the experiment, no liquid leakage was observed at the bottom of the container (since the liquid was dyed, any leakage would have been visible).

Regarding the issue of cross-contamination between the two phases you mentioned, in biphasic reactions involving water and low-polarity solvents (such as hexane and cyclohexane), the disparity in solubility between water and these solvents is indeed significant. The solubility of hexane in water is 0.014%, making cross-contamination between the phases negligible. We measured the relative content between the two phases, such as water and ethyl acetate (EtOAc), for systems where the two phases are slightly soluble. The content of EtOAc in water was determined by Total Organic Carbon (TOC) analysis, while the water content in EtOAc was measured using NMR.

The cellular reactor first adsorbs water and is then inserted into 50 mL of ethyl acetate after stirring at 50 rpm at room temperature. For 5 hours, the water phase is separated from the cellular reactor, yielding sample 1W, with the organic phase being sample 1EA. Using a commercially available stirrer, an amount of water equal to the holding capacity of the cellular reactor is added to

50 mL of ethyl acetate. After stirring at 50 rpm at room temperature for 5 hours, the water phase is extracted to obtain sample 2W, with the organic phase being sample 2EA. The process for obtaining sample 3 is identical to that of sample 2, except the stirring speed is changed from 50 rpm to 1000 rpm. After stirring and allowing the phases to separate, the water phase obtained is sample 3W, with the organic phase sample 3EA. The cellular reactor is made of resin material and could also be a potential organic carbon source. Therefore, we placed the cellular reactor in 50mL of water and stirred at 50 rpm at 60°C for 5 hours to verify whether the cellular reactor would contaminate the water phase, with the sample being labeled as 4W.

For the content of ethyl acetate in water, we approximate its concentration by measuring the Total Organic Carbon content using a TOC analyzer. The TOC analyzer employs a combustion method, oxidizing all organic substances to carbon dioxide, determining the organic carbon content.

Table S15 Total organic carbon in water

Sample	DI Water	S15-1W	S15-2W	S15-3W	S15-4W
TOC/ppm	2.13	1467	2425.5	1366	4.57

As a blank control, the TOC content in the pure deionized water used throughout the experiment was 2.13 ppm. Sample 4W showed a TOC content of 4.57 ppm for a second blank control, indicating that the cellular reactor released a minimal amount of organic substances into the water. This further demonstrates the stability of the cellular reactor in water. Under ambient temperature and stirring at 50 rpm, sample 1, obtained using the cellular reactor, had a lower ethyl acetate content in water compared to sample 2, obtained using a commercial stirrer, signifying that the use of the cellular reactor can significantly reduce cross-contamination between the two phases.

Nuclear magnetic resonance (NMR) spectroscopy provides the water content in ethyl acetate. The fundamental principle of quantitative NMR is as follows:

under the same test conditions, the peak area is directly proportional to the total number of hydrogen atoms corresponding to that peak. Therefore, the peak area is also proportional to the amount of substance and the equivalent number of hydrogen atoms. The calculation formula is as follows:

$$C_W = \frac{I_W/n_{WH}}{I_W/n_{WH} + I_{EA}/n_{EAH}}$$

In this context, C_W represents the mole fraction of water in ethyl acetate, I_W is the area of the peak corresponding to water, I_{EA} is the area of the peak corresponding to ethyl acetate, n_{WH} is the number of equivalent hydrogen atoms for the water peak, and n_{EAH} is the number of equivalent hydrogen atoms for the ethyl acetate peak. When quantifying using the hydrogen atoms on the methylene group of ethyl acetate, the formula is adjusted accordingly:

$$C_W = \frac{I_W/2}{I_W/2 + I_{EA}/2} = \frac{I_W}{I_W + I_{EA}}$$

The NMR data for samples S15-1EA to 3EA are as follows:

Figure S40 NMR of sample S15-1EA

Figure S41 NMR of sample S15-2EA

Figure S42 NMR of sample S15-3EA

Table S16 NMR of samples with water content in ethyl acetate

Sample	S15-1EA	S15-2 EA	S15-3 EA
I_W	656.87	607.38	923.58
I_{EA}	6800.13	6750.85	6809.8
C_W	8.8%	8.3%	12.1%

Under ambient conditions, the water content in ethyl acetate using the cellular reactor (S1 EA) is comparable to that without using the cellular reactor (S2 EA) but significantly lower than under vigorous stirring conditions (S3 EA). This indicates that while the cellular reactor increases the interfacial area, it does not introduce additional cross-contamination between the two phases. On the contrary, compared to vigorous stirring, the use of the cellular reactor actually reduces cross-contamination between the phases and facilitates rapid separation afterward, making it an advanced tool for chemical processes.

In the testing process for water content in ethyl acetate, samples were not diluted with deuterated reagents; instead, ethyl acetate was directly analyzed. Given that the water content in ethyl acetate is already minimal, dilution with deuterated reagents would increase the relative error in water content measurements due to the presence of small amounts of water in the deuterated reagents themselves. Therefore, deuterated reagents were not introduced in this part of the testing.

Additionally, after the reaction, apart from organic solvents, the water phase inside the cellular reactor may also contain a small amount of substrates/products that are slightly soluble in water. We conducted thermogravimetric analysis on the post-reaction aqueous phase to represent the degree of cross-contamination in the water phase accurately. Samples of the water phase from the acid-catalyzed addition of dihydropyran to alcohol and the base-catalyzed Knoevenagel condensation reaction were tested. The

testing was conducted in a nitrogen atmosphere, starting from room temperature and heating at a rate of 3 °C/min to 400°C.

Figure S43 Thermogravimetric curve of the aqueous phase after reaction

In the water sample test following acid catalysis (Fig S43a), the mass loss at 75.5°C is attributed to water, leaving 9.4% of the initial mass. Although this temperature does not reach the boiling point of water, the slow rate of temperature increase and the continuous flow of N₂ over the sample accelerated the loss of water. When the temperature reached 100°C, the remaining mass was 3.3% of the initial mass, indicating that all the water had been lost. At 337°C, the remaining mass was 1.9% of the initial mass, which should correspond to sulfuric acid. Therefore, the mass lost between 100-337°C originated from organic phase components, including substrates and products. This suggests that the degree of cross-contamination between the two phases is very low.

Three points of slope change were observed in the water sample test following base catalysis (Fig S43b). At the first change in slope, the temperature reached 44°C, with the remaining mass being 30.6% of the initial mass, indicating the beginning of the loss of ethyl acetate and water; at the second slope change, the temperature reached 80°C, with the remaining mass at 22.6% of the initial mass, marking the complete loss of ethyl acetate; at the third change in slope,

at 154 °C, the remaining mass was 9.7% of the initial mass, indicating the complete loss of water. When the temperature rose to 397 °C, the remaining mass was 5.73% of the initial mass. It suggests that the substrates, products, and any potential by-products dissolved in water account for less than 4% of the total mass of the aqueous phase.

The boiling points of all substances involved in the aforementioned process design are as follows in the table below.

Table S17: Boiling points of all substances in the aforementioned system

Compound	CAS No.	Boiling point
Cyclohexane	110-82-7	80.7 °C
3,4-Dihydro-2H-pyran	110-87-2	86 °C
Benzyl alcohol	100-51-6	205 °C
(Tetrahydro-2H-pyran-2-yl)benzyl ether	1927-62-4	~280 °C
Sulfuric acid	7664-93-9	337 °C
Ethyl acetate	141-78-6	77 °C
4-Chlorobenzaldehyde	104-88-1	214 °C
Malononitrile	109-77-3	220 °C
2-(4-Chlorobenzylidene)propanedinitrile	1867-38-5	~330 °C
Potassium carbonate	584-08-7	1689 °C (decomposes)

Therefore, the degree of cross-contamination between the two phases was determined through a combined analysis using various testing methods, including Total Organic Carbon (TOC) analysis, NMR, and thermogravimetric analysis. For partially miscible systems (water and ethyl acetate), the degree of cross-contamination between the phases was similar, whether using the cellular reactor or not. However, the cellular reactor's ability to restrict fluid movement and simplify the subsequent separation of water and oil phases is an advantage that conventional stirring systems cannot offer. Hence, we

believe that incorporating the cellular reactor into biphasic liquid-liquid reactions is meaningful.

Q4 I am not an expert of simulation and modeling, but I do not really think that the paragraph "Modeling of interfacial reactions" is needed for this work. Also, the proposed model is valid for a flat interface between static fluids, which does not correspond to the experimental situation.

Response: Thanks very much for your valuable feedback. We have taken your advice and removed this analysis from the main text. Our initial intention for conducting interface reaction modeling and MD simulations was to facilitate a more comprehensive understanding of the reaction processes occurring at interfaces. We have moved this section to the Supporting Information for interested readers to consult.

Q5 further insight is needed to understand the benefits of using cellular reactors for chemical transformations. For example, comparison of reaction outcomes at varying rotation speeds might offer a starting point to evaluate the performances of the cellular reactor with respect to a standard bladed impeller. Reproducibility of reactions outcome might be a good parameter to evaluate as well (I would expect better reproducibility by using the cellular reactor).

Response: Thanks for your valuable suggestion. First, we discuss the benefits of using a cellular reactor for chemical reactions. We conducted chemical reactions at various stirring speeds using the cellular reactor, taking the acid-catalyzed addition reaction of dihydropyran to alcohol as a typical example. The conditions included were static (0 rpm), and at 50, 100, 150, and 200 rpm.

Table S18 Conversion of dihydropyrans at different rotational speeds

Time (min)	Conversion rate (%)				
	0 rpm	50 rpm	100 rpm	150 rpm	200 rpm
30	0.0	6.4	5.1	8.3	8.0
45	1.6	9.2	5.3	8.2	10.0
60	0.7	16.4	12.4	17.5	31.2
90	6.2	20.3	23.1	24.5	49.5
120	26.9	36.8	33.5	33.6	60.9
150	12.1	42.7	45.5	41.2	70.8

It is evident that the reaction rate is very slow in the absence of stirring. At a stirring speed of 200 rpm, liquid leakage occurs within the cellular reactor. This is due to insufficient adhesive forces to counterbalance the increased centrifugal force at high stirring speeds. The reaction rates are relatively similar at lower stirring speeds (50, 100, and 150 rpm). The impact of stirring speed on reaction rate is quite limited at these speeds. It is because increasing the stirring speed primarily enhances the mass transfer rate of substances in the external organic phase, while the rate-determining step of the interface reaction involves the diffusion of substrates and catalyst components to the interface, not just the mass transfer within the external oil phase. Therefore, increasing the stirring speed does not significantly enhance the reaction rate.

Table S4 Parameters of cellular reactors in this study

Ractor	Structure	Location	L	D*	Blades	N
Ractor-A		Fig. 3e	2 mm	0.33	3	498
Ractor-B		Fig. 3e	2 mm	0.33	4	568

Ractor-C		Fig. 3e	2 mm	0.33	6	708
Ractor-D			2 mm	0.33	6	708
Ractor-E			2 mm	0.33	6	708
Ractor-F			-	-	6	-
Ractor-G			-	-	-	-
Ractor-H			-	-	-	-

Table S7 Comparison of the solid cellular reactor with two types of commercially available reactors

Time (min)	Conversion rate (%)		
	Ractor-F	Ractor-G	Ractor-H
30	1.9	4.1	3.8
60	2.5	6.9	8.1
90	4.9	8.6	10.2

Dihydropyran (12.5 mmol) and benzyl alcohol (12.5 mmol, 1.0 equiv) were combined in 50 mL of cyclohexane. An equivalent volume of 1 M H₂SO₄ solution, matching the liquid holding capacity of the cellular reactor, was added into the cellular reactor and inserted into the organic phase, stirring at 50 rpm. The

reactions were conducted in custom-made three-neck flasks heated to 60 °C.

It is evident that, whether employing a solid stirring paddle without cell units or a commercially available generic stirring paddle, the reaction rate is significantly slower than when using the cellular reactor. Experimental observations indicate that the design and use of the cellular reactor are necessary.

Q6 further insight is also needed to understand the problems/limitations related to the use of cellular reactors for chemical transformations. For example, can this reactor be used to perform reactions leading to products which partition in both phases? Can this reactor be used to perform reactions sensitive to oxygen or carbon dioxide? What problems arise in case of formation of an insoluble product/byproduct (that therefore crushes out from solution)?

Response: Thanks the reviewer for the valuable comments. In this paper, our primary goal is to present the design concept, principles, manufacturing process, and basic applications of the reactor in water-oil biphasic systems. The issues you raised can indeed occur in actual reactions. For instance, the distribution of products between water and oil phases does exist in some reactions, but in practical operations, both in laboratory and industrial processes, efforts are made to avoid this situation by adjusting the solvent system or reactant composition, as it significantly increases the cost and energy consumption for subsequent material separation. This type of reactor can also be used in reactions involving oxygen and carbon dioxide. However, such reactions involve gas-liquid-solid three-phase processes, including the transfer and diffusion of gases in liquids and the contact of gases at solid-liquid interfaces. The physical equations and design concepts used in our work would need to be adjusted accordingly, which is beyond the scope of this study. However, your suggestion has inspired us to consider new research focusing specifically on gas-liquid-solid three-phase reaction systems. Regarding the precipitation of insoluble products in reactions, we conducted experiments

using hollow glass microbeads as analogs for insoluble products.

Hollow glass microspheres were purchased from 3M Company, with sizes ranging from 9-25 μm and a density of 0.6 g/cc. They were added at a concentration of 0.01 wt% to 50 mL to simulate solid impurities generated in certain reactions. The reaction rate of reactions with added hollow glass microspheres was very close to those without the microspheres. Therefore, the cellular reactor demonstrates good applicability for reactions that generate precipitate as a by-product.

Q7 As a final remark, the manuscript needs some improvement in terms of clarity, and referencing. At the first read, I had some difficulties in understanding the meaning of reactor parameters L and D as listed in the text, later I found them in figure 2a, but they are not obvious to see (and there is no direct reference in the text to their presence in figure 2a). Maybe changing the color code in order to have all the reactors trusses of the same color in all the four 2a pictures would help. Also, parameters φ (page 7, line 3) and θ (page 8, line 2) are not defined. In figure 2d, parameter G^* is present, but is not defined anywhere in the text. In figure 3, graph c is entirely dedicated to the dependence of parameter λ from D^* , however λ is not clearly defined in the text (it is only presented as a dimensionless number describing the wetting properties of the cell). After description of cellular reactors geometries SC, BCC and BCC with frame, there is no clear indication of which geometry has been used to run the reactions. Finally, when referring to the reusability of the cellular reactor (page 21, line 10), the authors should state in the text what is the reaction they are re-running (it is said in the SI but not in the main text), especially because it is none of the thioether oxidations described in the previous paragraph. Regarding the referencing, very little space is dedicated to the description of the state of the art in literature, the authors should give a broader overview of the field (not only related to Pickering emulsions, e.g. see

10.1021/acsnano.8b05718, 10.1021/acsami.8b14321, 10.1038/s41598-019-42769-8). Also, ref. 6 is placed incorrectly.

Response: Thanks the reviewer for the valuable comments, and we apologize for not clearly explaining these issues in the article.

Figure 1 (a) Schematic structure of syncytium and cellular reactor (b) The three cell unit types used in this study and a schematic of the cross sections

Regarding the definitions of L and D , we have indicated in the text that their definitions are provided in Figure 1b.

P6 Line12:

where D represents the diameter of the cell truss, L is the cell length, and D^* is a characteristic dimension of the cell (Fig.1b)

The parameters ϕ and θ were initially defined in the Supporting Information (SI), and we did not mention them in the main text; their definitions have now been added to the manuscript.

P6 Line13:

ϕ is the ratio of truss volume to total volume in a single cell.

P9 Line2:

θ denotes the three-phase contact angle between liquid 1 and 2 and the solid.

G^* is represented as the normalized gravity, denoted in the formulas as F_g^* .

We have now standardized all symbols throughout the text.

P9 Line6:

$$F_g^* = \rho g \varepsilon$$

The relative characteristic capillary force versus the characteristic gravitational force magnitude reflects how well the cell holds liquid (it is worth noting that for BCC and BCC+frame, porosity ε exists when $D^* < 0.8$)

Since both characteristic gravity and characteristic capillary forces are positive values, we opt to take the logarithm of their ratio for plotting. This approach not only eliminates dimensions but also compresses the range of the vertical axis for easier viewing. Furthermore, we also plot the logarithm of the ratio of characteristic gravity to itself in the chart, essentially adding a $y=0$ line to the graph. Above this line, the characteristic capillary force is greater than the characteristic gravity, indicating that a cellular reactor satisfying this condition can stably hold liquid. Conversely, below this line, the reactor would leak liquid.

Regarding the equivalent wetting coefficient λ , it is defined as a dimensionless

number describing wettability characteristics. We have verified the working curve of λ through a process involving 3D modeling, data extraction, and curve fitting, as outlined below:

Step 1: We utilized the 3D modeling software Rhino to construct models of unit cells with different types and varying edge lengths ($L=1$), facilitating subsequent normalization operations.

Step 2: The area (A_{SL1}) in contact with the inner liquid phase was extracted as the working point.

Step 3: Using Origin software, we performed curve fitting based on the obtained data to derive the working curve of λ .

The type of cellular reactor used in the paper is mentioned in the Methods section of the main text; please refer to P22 Line 20

Furthermore, we have added information about the reactors used in each reaction description to facilitate readers' understanding of the experimental details. For photos and detailed information about the cellular reactor, refer to Table S4

the lower limit of the range of values for the equivalent diameter, D_{min}^* , as a function of D^*

Additionally, the operating point of the cellular reactor is also added in Figure 3d (indicated by the asterisk in the figure representing the working point of the cellular reactor).

Regarding the unclear description of the cyclic reactions in the main text, we have added an explanation in the main text to ensure that readers can directly understand what kind of reactions were conducted.

P21 Line 6:

The reactor was used to repeatedly perform the acid-catalyzed addition reaction (Entry 1) five times, without any decrease in reaction yield (Table S14)

We have carefully reviewed the literature you mentioned: 10.1021/acsami.8b14321, 10.1021/acsnano.8b05718, 10.1038/s41598-019-42769-8. The research conducted in these papers is commendable, covering many aspects we had not considered. Therefore, we have drawn upon the strengths of the aforementioned publications and cited them in our article, hoping to enrich our study. The positions of the aforementioned three references correspond to citations 5, 15, and 4 in the manuscript, respectively. Reference 6 in the original manuscript has been deleted.

Reviewer: 3

Overall Comments:

This is an exciting piece of work demonstrating cellular fluidics as a method to perform liquid-liquid reactions. Cellular fluidics has the potential to allow for enhanced interfacial area between the two liquid phases, while maintaining continuous liquid phases (ie no emulsions). In their work, the aqueous phase contains a catalyst or oxidant and is contained within the inert cellular fluidic structure and submerged in an organic phase which contains the substrate. An analysis was performed to predict geometries which would function as intended. A visual demonstration using a pH indicator was used to show the reaction occurring in the cellular fluidic reactor. They demonstrate this new format for

liquid-liquid reactions with four different reactions and quantified them with GC-MS.

This work builds on 3D printed structures to order two immiscible fluids for liquid-liquid reactions. While these concepts are not completely novel, this work is significant in demonstrating how these concepts can be used for chemical reactions. Furthermore, the analysis presented suggests a relatively straightforward method to determine if a geometry is stable and can be used for this function.

Response: We appreciate the reviewer's highly positive comments. We tried our best to address all the detailed comments and made a detailed revision of our manuscript.

Q1 More background and comparison to liquid-liquid reactors. I understand if this work does not perform well compared to the state of the art liquid-liquid reactors, but I would still like to see some comparison and discussion of where this technology can be further improved. How does the time scale for this work compare to the state-of-the-art analytical chemistry tools for liquid-liquid extraction where reactions are conducted in glass tubes and that corrosion is not a concern?

Response: Thanks the reviewer for the valuable comments, More background and information on liquid-liquid reactions have been added to the manuscript, P2 Line 14. The content is as follows

Liquid-liquid reactions are vital in chemical synthesis due to the fast mass transfer between interfaces, uniform temperature field distribution, dynamic interfacial composition updating, and resistance to catalyst poisoning. Typically, one reactant acts as a dispersed phase, forming tiny droplets in the continuous phase of another substance, increasing the area for molecular diffusion. The surface tension causes the dispersed-phase droplets to adopt a spherical shape, minimizing the system's energy. However, collisions between droplets lead to reduced contact between the liquid phases, often necessitating fast

mechanical stirring or emulsifiers.

Stirred reactors are widely used in chemical engineering and industrial processes for liquid-liquid reactions. Stirring facilitates thorough mixing of the two phases, enhancing reaction rates. It also helps maintain temperature uniformity, preventing temperature gradients due to exothermic or endothermic reactions. However, stirred reactors have drawbacks, such as high shear forces that can damage shear-sensitive chemicals, and insufficient stirring speeds that fail to maintain adequate interfacial area between the two liquid phases, leading to reduced reaction rates.

Using emulsifiers stabilizes droplets and reduces their size, thereby increasing the total interfacial area. However, the compatibility of emulsifiers with the reaction system must be considered, as the introduction of inappropriate emulsifiers can decrease reaction efficiency. Additionally, the resulting emulsion complicates the separation and purification of products post-reaction. Most importantly, the use of emulsifiers inevitably requires rapid mechanical stirring.

Phase transfer catalysts can enhance the efficiency of liquid-liquid biphasic reactions without the need for vigorous stirring. However, the subsequent separation process becomes more complex due to the high affinity of phase transfer catalysts for both phases. Therefore, designing a novel reactor that can enhance liquid-liquid interfacial mass transfer at low stirring speeds and allow for easy separation after the reaction is crucial.

In nature, numerous liquid-liquid reactions exhibit remarkable controllability. Cell membranes represent a widely occurring natural interface for such reactions, where reactants diffuse to the interface, and products depart, maintaining interface stability throughout the process. Syncytia, large cellular structures formed by the fusion of multiple cells, facilitate intercellular cooperation and functional integration, enhancing cellular stability and efficiency. Inspired by the structure of syncytia and combining this with the structure of stirred reactors, we propose a novel type of cellular reactor. This

design merges multiple units into a single entity, enhancing strength and centralizing functions.

We also consulted the literature and identified a study involving acid-catalyzed addition reaction of dihydrofuran and benzyl alcohol in liquid-liquid reactions, which we compared with our findings.

In terms of catalytic efficiency, a comparative experiment was conducted using 0.1 M H₂SO₄. The reaction volume was 50 mL, with 12.5 mmol of dihydrofuran. At a reaction time of 300 min and a heating temperature of 60°C, the conversion rate reached 97%. In the referenced study, continuous synthesis was carried out using 0.1 M H₂SO₄ with a dihydrofuran concentration of 0.25 M. The reaction was conducted for 500 h at 50°C with a flow rate of 0.6 mL/h, maintaining a conversion rate of 81%. 75 mmol of dihydrofuran was reacted at an 81% conversion rate after 500 h. In contrast, in our work, 75 mmol of dihydrofuran was converted in just 30 h with a yield of 97%.

Comparison of separation difficulties: A Pickering emulsion system was employed in the referenced study, requiring pre-reaction emulsification and post-reaction centrifugation for catalyst recovery. In contrast, this research utilized a cell reactor, confining the catalytic components within it. Before the reaction, the catalyst (sulfuric acid) was simply adsorbed onto the reactor, and post-reaction separation involved lifting the cell reactor from the reaction apparatus. If catalyst liquid recovery was necessary, it could be achieved by centrifuging the liquid out into a beaker for collection.

Our comparison is limited to these two aspects. The referenced work focused on creating stable Pickering emulsions for catalytic reactions, while the innovation in this study lies in establishing a stable liquid-liquid interface and facilitating a straightforward and efficient separation process during the reaction.

Reference: <https://doi.org/10.1021/jacs.6b04265>,

We have also contemplated the primary limitation of cell reactors, which lies in

the relatively limited amplification of the interface (only 2.3 times). According to computational data, the increase in interface area achieved by the cell reactor is only 2.3 times compared to direct liquid addition. While this situation still yields satisfactory catalytic efficiency, further enhancement of the interface area could potentially accelerate the chemical reaction rate to a certain extent. Possible improvements include 1) Optimization at the basic unit level, i.e., designing more efficient cell units to increase the interfacial area at the unit level. 2) Optimizing the arrangement logic, as the current cell units are arranged in the simplest close-packing manner. Following the approach outlined in the literature, larger structures can be inserted in the middle of smaller cells. With dimensions exceeding capillary lengths, the catalytic liquid will not be adsorbed in these larger structures, creating gaps in the middle of the cells. This allows the reaction liquid to pass through during the rotation process, thereby increasing the interfacial area between the two phases. 3) Building on point 2, manufacturing a cell reactor using two types of cell units. This involves interweaving conventional cells with units that do not adhere strictly to the liquid-holding rule, achieving effects similar to point 2.

We employed a cell reactor for liquid-liquid extraction processes. Riboflavin, also known as Vitamin B2, is an essential nutrient for the human body. Initially, riboflavin dissolves in ethyl acetate, imparting a pale yellow color. A 1M NaOH aqueous phase is adsorbed into the cell reactor. A stirring paddle is inserted into the organic phase and fixed, followed by rotation at a speed of 50 rpm. As stirring progresses, the pale yellow color in the solution gradually lightens, indicating the transfer of riboflavin molecules from the organic phase to the aqueous phase, completing the extraction process.

Using UV-Visible spectroscopy, we measured the riboflavin concentration in the organic phase before and after extraction. The extraction efficiency was calculated as 58.9% (concentration-based) based on the following formula.

$$\eta = \frac{C_0 - C_1}{C_0} \times 100\%$$

In this context, η represents the extraction efficiency, where C_0 denotes the initial concentration and C_1 represents the concentration in the residual phase after extraction.

Compared with state-of-the-art extraction tools, we also reviewed the literature on riboflavin extraction. Riboflavin was successfully extracted using lactic acid ethyl ester and organic salt-based ATPS in the system of lactic acid ethyl ester (1) + sodium citrate (2) + water (3), achieving a maximum efficiency of 87.6%. While the extraction efficiency of the cell reactor falls short of previously reported extraction tools, we believe that through modification and optimization, there is potential for expanded applications and improved efficiency in the future. Reference:DOI: 10.1021/acs.jced.1c00909

Figure S14 Photographs and SEM images of the cellular reactor before and after the cycling reaction (a-d) raw state, not processed in any way (e-h) after cyclic reactions

It is observable that after five cycles, the cellular reactor underwent minimal changes in appearance (Fig S14a, e). The overall structure remained intact (Fig

S14b, f), with only slight fine lines appearing on the surface microstructure (Fig S14c, d, g, h). The degree of corrosion observed is entirely acceptable, considering the number of cycles it underwent.

Even after five cycles of use, the cellular reactor remained in excellent condition, preserving its capacity to facilitate liquid-liquid two-phase interface formation. This led to a notable increase in the cellular reactor's utilization efficiency and improved the reaction's economic feasibility.

Q2 The analysis for eq 1-7 look similar to (22) however, equations 4-7 look different. It is unclear if the adhesive force is normalized? The units do not look correct for eq4 and 6. I am unclear why the surface-liquid area is being integrated over instead of the contact perimeter?

Response: Thanks the reviewer for the valuable comments. As you correctly identified, there were errors in the forms of Equations 4 and 6. The correct equations have now been revised and re-introduced into the main text. We have also carefully verified the units of the new formulas.

For a more intuitive comparison between adhesive force and gravity, the adhesive force has been simplified in the same manner as gravity without undergoing true normalization. We believe this approach will help readers clearly and concisely determine how to construct a cellular reactor that meets the requirements.

The relative characteristic capillary force versus the characteristic gravitational force magnitude reflects how well the cell holds liquid (it is worth noting that for BCC and BCC+frame, porosity ϵ exists when $D^* < 0.8$)

Since both characteristic gravity and characteristic capillary forces are positive values, we opt to take the logarithm of their ratio for plotting. This approach not only eliminates dimensions but also compresses the range of the vertical axis for easier viewing. Furthermore, we also plot the logarithm of the ratio of characteristic gravity to itself in the chart, essentially adding a $y=0$ line to the graph. Above this line, the characteristic capillary force is greater than the characteristic gravity, indicating that a cellular reactor satisfying this condition can stably hold liquid. Conversely, below this line, the reactor would leak liquid.

Q3 I believe there is an error in eq9 for the SI

Response: Thanks the reviewer for the valuable comments. Indeed, Equation 9 in the Supporting Information should be in this form.

$$A_s = A_{SL1} + A_{SL2}$$

Q4 I could not follow the analysis after table S1 in the SI. The surface tension data for water and diiomethane is presented in the table, but the analysis is for water and cyclohexane? Its unclear what values are measured and which ones are derived. Plugging in the numbers seems to give different numbers than what is presented. I think it would benefit from a table presenting all the measured values for the different systems used in this work. This may also help clarify the different.

Response: Thanks the reviewer for the valuable comments. We have revised the analysis process of surface tension in SI units. Subscripts for various symbols have been tabulated to enhance readers' comprehension of the physical meanings of each symbol. The modified text is as follows:

Figure S3 Schematic diagram of the three-phase contact angle between liquid 1, liquid 2 and solid

The procedure for measuring and calculating the three-phase contact angle of the cellular reactor truss with water and cyclohexane was as follows, with the test environment at 20°C. First the solid truss surface tension was measured. The solid surface tension can be regarded as the sum of dispersive surface tension and polar surface tension:

$$\sigma_s = \sigma_s^d + \sigma_s^p \quad (1)$$

where σ_s is the surface tension of the solid, σ_s^d is the dispersive surface tension of the solid, and σ_s^p is the polar surface tension of the solid.

And the interfacial tension of solid-liquid can be expressed as

$$\sigma_{sl} = \sigma_s + \sigma_l - 2\sqrt{\sigma_s^d \sigma_l^d} - 2\sqrt{\sigma_s^p \sigma_l^p} \quad (2)$$

where σ_{sl} represents the interfacial tension of the solid-liquid, σ_l represents the surface tension of the liquid, σ_l^d represents the dispersive surface tension of the liquid, and σ_l^p represents the polar surface tension of the liquid.

The Young equation for the solid-liquid-gas three-phase is given by:

$$\sigma_s = \sigma_{sl} + \sigma_l \cos \theta \quad (3)$$

Substituting equations (1) and (2) into equation (3), we have

$$(1 + \cos \theta)\sigma_l = 2\sqrt{\sigma_s^d \sigma_l^d} + 2\sqrt{\sigma_s^p \sigma_l^p} \quad (4)$$

The surface tension of a solid can be determined by utilizing just two liquids with known dispersive and polar surface tensions, exhibiting a significant polarity difference. Water and diiodomethane were chosen as test liquids for

measuring truss surface tension because their dispersive and polar surface tensions are known.

$$(1 + \cos \theta_W)\sigma_{lW} = 2\sqrt{\sigma_s^d \sigma_{lW}^d} + 2\sqrt{\sigma_s^p \sigma_{lW}^p} \quad (5)$$

$$(1 + \cos \theta_D)\sigma_{lD} = 2\sqrt{\sigma_s^d \sigma_{lD}^d} + 2\sqrt{\sigma_s^p \sigma_{lD}^p} \quad (6)$$

where l_W represent water, l_D represent diiodomethane and s represent the reactor truss materials. $\cos \theta_W$ represent the contact angle of water-air-solid truss materials and $\cos \theta_D$ represent the contact angle of diiodomethane-air-solid truss materials.

Table S1 Surface tension data of water and diiodomethane at 20°C

Liquid	σ_l (mN/m)	σ_l^d (mN/m)	σ_l^p (mN/m)
l_W	72.8	21.8	51.0
l_D	50.8	49.5	1.3

Calculate the surface tension of the truss material

$$\sigma_s = \sigma_s^d + \sigma_s^p = 62.46 \text{ mN/m}$$

Interfacial tension of water-truss material

$$\begin{aligned} \sigma_{slW} &= \sigma_s - \sigma_{lW} \cos \theta_W = 8.78 \text{ mN/m} \\ \cos \theta_W &= 33.2^\circ \end{aligned}$$

Interfacial tension of cyclohexane-truss material

$$\begin{aligned} \sigma_{slC} &= \sigma_s - \sigma_{lC} \cos \theta_C = 36.13 \text{ mN/m} \\ \cos \theta_C &= 0^\circ \end{aligned}$$

where l_C represent cyclohexane, and $\cos \theta_C$ represent the contact angle of cyclohexane-air-solid truss materials.

Interfacial tension of water-cyclohexane

$$\sigma_{lWlC} = 44.35 \text{ mN/m}$$

According to the three-phase Young's equation of water-cyclohexane-truss

material, the three-phase contact angle is obtained.

$$\sigma_{sIW} = \sigma_{sIC} + \sigma_{IWIC} \cos \theta$$

$$\theta = 52.55^\circ$$

where $\cos \theta$ represent the contact angle of water-cyclohexane-solid truss material.

In the above process, only the dispersive surface tension and polar surface tension of water and diiodomethane were obtained from the literature, and the remaining surface tension and contact angle data were determined experimentally or calculated based on the measured data according to the formula.

Table Sx Description of interfacial tension

	σ_{IW}	σ_{ID}	σ_s	σ_{IWIC}	σ_{sIC}	σ_{sIW}
Description	gas-liquid	gas-liquid	gas-solid	liquid-liquid	solid-liquid	solid-liquid
Source	measured	measured	calculated	measured	calculated	calculated

Where *measured* denotes that it is derived from direct measurements or derived entirely from measured data. *Calculated* denotes that part of the data is derived from the literature and combined with measured data.

Table Sx Description of inter contact angle

Contact angle	θ	$\cos \theta_W$	$\cos \theta_C$	$\cos \theta_D$
Description	iquid-liquid-solid	gas-liquid-solid	gas-liquid-solid	gas-liquid-solid
Source	calculated	measured	measured	measured

Where *measured* denotes that it is derived from direct measurements or derived entirely from measured data. *Calculated* denotes that part of the data is derived from the literature and combined with measured data.

Q5 The equivalent wetting factor, λ , suggests an exciting simple method to

evaluate different geometries for their ability to hold fluid. Yet, there is little to no description of how this parameter is calculated or measured. Please provide additional information and corresponding experiments (?) necessary to determine the wetting factor.

Response: Thanks the reviewer for the valuable comments. The concept of the equivalent wetting coefficient is introduced as a dimensionless number to quantify various cellular structural forms.

Q6 The molecular dynamics simulations are difficult to follow and weakens this work. This takes away from the paper and opens more questions. For example, the size of the simulation (number of atoms, dimensions of cell) is not specified. The time step units are not specified. It is not clear what the density is normalized by. Fig 4b is misleading- its unclear what this shows. This analysis should be removed.

Response: Thanks the reviewer for the valuable comments. The initial purpose of employing molecular dynamics simulations was to facilitate the understanding and construction of a model for interfacial reactions. However, providing insufficient information has led to confusion. We have subsequently removed this analysis from the main text and relocated it to the supporting information for the perusal of interested readers, thereby avoiding further reader perplexity.

Q7 A key feature that is barely discussed is the rotation rate. The rotation rate is stated in the methods but there is no discussion about the effect of the rotation, the maximum rate allowable to keep the fluid contained, or even a sentence suggesting a limit to the rotation rate. While I understand if an extensive analysis of the rotation rate is outside the scope, simply calculating the capillary number, and including a discussion of how the rotation rate affects the operation is necessary. What are the reaction rates with no rotation? With rotation?

Response: Thanks the reviewer for the valuable comments. We configured the

parent structure of the cellular reactor as a stirrer shape, with the primary objectives being an increase in the contact area between the inner and outer phase liquids and the capability for rotation. Rotation serves a vital function by reducing concentration gradients in the outer phase liquid and facilitating the transfer of products from the interface to the outer phase.

As the rate-limiting step across the interfacial chemical reaction is diffusion, meaning the reaction rate is determined by the rate at which molecules diffuse from the bulk phase to the interface, we opted for a low rotation speed (50 rpm). Given the relatively slow reaction rate, mild rotation is sufficient to eliminate external diffusion. Furthermore, at low rotational speeds, the liquid-liquid interface can be approximated as quasi-static, contributing to the effective diffusion of molecules to this region.

We also conducted catalytic reaction tests at different rotation speeds. Excessive rotation speeds resulted in substantial centrifugal forces, leading to leakage of the inner phase liquid from the cellular reactor. The critical threshold was approximately 200 rpm. Therefore, investigating rotation speeds exceeding 200 rpm was deemed inconsequential.

Table S18 Conversion of dihydropyrans at different rotational speeds

Time (min)	Conversion rate (%)				
	0 rpm	50 rpm	100 rpm	150 rpm	200 rpm
30	0.0	6.4	5.1	8.3	8.0
45	1.6	9.2	5.3	8.2	10.0
60	0.7	16.4	12.4	17.5	31.2
90	6.2	20.3	23.1	24.5	49.5
120	26.9	36.8	33.5	33.6	60.9
150	12.1	42.7	45.5	41.2	70.8

It is evident that the reaction rate is very slow in the absence of stirring. At a

stirring speed of 200 rpm, liquid leakage occurs within the cellular reactor. This is due to the adhesive forces being insufficient to counterbalance the increased centrifugal force at high stirring speeds. At lower stirring speeds (50, 100, and 150 rpm), the reaction rates are relatively similar. At these speeds, the impact of stirring speed on reaction rate is quite limited. This is because increasing the stirring speed primarily enhances the mass transfer rate of substances in the external organic phase, while the rate-determining step of the interface reaction involves the diffusion of substrates and catalyst components to the interface, not just the mass transfer within the external oil phase. Therefore, increasing the stirring speed does not significantly enhance the reaction rate. This is because the rate-determining step of this interfacial reaction involves the diffusion of substrates and catalytic components to the interface rather than chemical reaction kinetics.

In catalytic reactions conducted at different rotational speeds (50, 100, and 150 rpm), remarkably similar reaction rates were observed. This indicates that variations in rotational speed did not significantly influence the reaction rate, confirming that the rate-determining step is not the interfacial reaction. This supports the accuracy of our initial hypothesis.

Q8 Another highlight of this technology is being able to separate and collect the different phases after the reaction. There is very little discussion of how liquid is recovered from the cellular fluidic device. If possible a video of this would be incredibly informative, otherwise a more detailed description is warranted. What is the recovery rate?

Response: Thanks the reviewer for the valuable comments. After the completion of the reaction, separating the inner and outer phase liquids is a straightforward process. The outer phase liquid can be obtained as a product solution without requiring additional steps. Separating the inner phase liquid poses a higher challenge due to the strong interactions with the cellular reactor. However, we employed centrifugation to collect the inner phase liquid. This

involved placing the entire cellular reactor inside an empty beaker and increasing the stirrer's rotational speed, such as to 300 rpm. Under the influence of centrifugal force, the liquid inside the cellular reactor separated and was collected in the beaker.

Q9 I would really like to see more comparison with experiment to validate Fig 3. By modifying the unit cell size I think it may be possible to print structures above and below the $y=x$ line (ie where the device should and should not work). Plotting 3-5 experimental points on figure 3d would be very helpful. Where is the device you used? A single unit cell, or a small group of cells would be sufficient for this demonstration and allow you to probe different unit cell types.

Response: Thanks the reviewer for the valuable comments. Following your guidance, we have designed and fabricated a range of different cells to further demonstrate liquid-holding stability. The design parameters for each cell are as follows.

Table S11 Parameters of cellular reactors

Sample	Cellular type	L	D*	Liquid state
S13-a	Cubic	4	0.2	Leaked
S13-b	Cubic	3	0.2	Leaked
S13-c	BCC	4	0.333	Held
S13-d	BCC	3	0.333	Held
S13-e	BCC	2	0.333	Held
S13-f	BCC+frame	4	0.333	Held
S13-g	BCC+frame	3	0.333	Held
S13-h	BCC+frame	2	0.333	Held

Figure S13 Liquid state demonstration of different cellular reactors

Figure S13 demonstrates the liquid retention effect. For cells S13-a&b, since they do not satisfy the formula $(\pi \frac{\lambda}{n} \cos \theta + 1)^{-1} < D^* < 1$, their operating points in Figure 3d, at $n=2,3d$, are located above the $y=x$ line, and for $n=1$, very close to the $y=x$ line, resulting in liquid leakage within the cells. However, for cells

S13-c-h, the operating points are below the $y=x$ line, thereby effectively retaining the liquid. In this case, the liquid is water, dyed with methylene blue.

Furthermore, the operating points of the aforementioned cells have been incorporated into the new Figure 3d. In this figure, asterisks denote the working points used throughout the study, circles represent liquid retention, and crosses indicate liquid leakage.

the lower limit of the range of values for the equivalent diameter, D_{min}^* , as a function of D^*

Q10 Colorimetric demonstration: Video 1: There looks like there is bubble formation on the left blade for the first half of the video. Fig 4a: Is there an explanation for why the top part the blade turned blue while the bottom stayed clear for the first 60s of the reaction in Fig 4a? Is there displacement of fluid?

Response: We appreciate your valuable insights. After closely observing the video and the actual experimental process, we conclude that the bubbles formed are likely due to a small amount of gas adsorbed on the surface of the stirrer. These trace bubbles did not have any impact on the experimental results. The cellular reactor is connected to the stirrer, and the stirrer shaft is secured to prevent any sliding displacement with screws. The colorimetric reaction remains static. The slight oscillation observed in the video is a result of the tightening of the screws. It is possible that the cellular reactor, having absorbed

some water solution in the air before immersion in n-hexane, carried a portion of gas into the system, resulting in bubble formation. Alternatively, gases dissolved in both phases may have concentrated at the interface, leading to bubble formation. We believe that pre-degassing the cellular reactor before the experiment might help improve this aspect.

Regarding your observation about the upper half of the reactor blades turning blue before the lower half within the initial 60 seconds, there is no issue related to liquid replacement. The experimental video was recorded continuously. We attribute this phenomenon to the disturbance caused by the insertion of the stirrer from top to bottom into the liquid, which led to convective motion in the upper portion of the container. Consequently, material transfer due to convection was stronger in the upper part of the liquid. This resulted in more reactants being transferred to the liquid-liquid interface in the upper half, causing it to change color first, followed by the lower half.

Q11 Other concerns and comments

The amount of text (and figure 1) dedicated to drawing parallels between biological systems and this device seems excessive. While I am not opposed to the narrative of being inspired by natural systems, the detailed description of cells, syncytium etc seems unnecessary. Especially since each unit cell is identical in the reactors, it's difficult to think of each of these bionic cells as individual cell types contributing to the whole organism. I was left wanting much more background about chemical liquid-liquid reactors, the current state of the art and the gold standard for performing these types of reactions. I suggest toning down the analogy, and instead going into more detail about liquid-liquid reactors. I would also try to change figure 1 to add more value- perhaps moving something like fig 2a into fig1- or having a general schematic of the different oil/aq phase and inert structure? A schematic of the reactions taking place? The overall concept and experiment was difficult to understand for people not familiar with 3D printed cellular reactors, a visual clearly describing the different

phases (organic, aqueous) and the reaction occurring would be very helpful.

Response: We appreciate your feedback, which is of utmost importance to us. We have realized that dedicating substantial space to describe the behavior inspired by our initial insights is inappropriate when outlining the fundamental concept of the liquid-liquid reactor. Therefore, we have reduced the description of biological membrane behavior and coacervate formation, opting for a brief mention. Our focus is now primarily on the liquid-liquid reactor. The revised background section is as follows:

Liquid-liquid reactions are vital in chemical synthesis due to the fast mass transfer between interfaces, uniform temperature field distribution, dynamic interfacial composition updating, and resistance to catalyst poisoning. Typically, one reactant acts as a dispersed phase, forming tiny droplets in the continuous phase of another substance, increasing the area for molecular diffusion. The surface tension causes the dispersed-phase droplets to adopt a spherical shape, minimizing the system's energy. However, collisions between droplets lead to reduced contact between the liquid phases, often necessitating fast mechanical stirring or emulsifiers.

Stirred reactors are widely used in chemical engineering and industrial processes for liquid-liquid reactions. Stirring facilitates thorough mixing of the two phases, enhancing reaction rates. It also helps maintain temperature uniformity, preventing temperature gradients due to exothermic or endothermic reactions. However, stirred reactors have drawbacks, such as high shear forces that can damage shear-sensitive chemicals, and insufficient stirring speeds that fail to maintain adequate interfacial area between the two liquid phases, leading to reduced reaction rates.

Using emulsifiers stabilizes droplets and reduces their size, thereby increasing the total interfacial area. However, the compatibility of emulsifiers with the reaction system must be considered, as the introduction of inappropriate emulsifiers can decrease reaction efficiency. Additionally, the resulting

emulsion complicates the separation and purification of products post-reaction. Most importantly, the use of emulsifiers inevitably requires rapid mechanical stirring.

Phase transfer catalysts can enhance the efficiency of liquid-liquid biphasic reactions without the need for vigorous stirring. However, the subsequent separation process becomes more complex due to the high affinity of phase transfer catalysts for both phases. Therefore, designing a novel reactor that can enhance liquid-liquid interfacial mass transfer at low stirring speeds and allow for easy separation after the reaction is crucial.

In nature, numerous liquid-liquid reactions exhibit remarkable controllability. Cell membranes represent a widely occurring natural interface for such reactions, where reactants diffuse to the interface, and products depart, maintaining interface stability throughout the process. Syncytia, large cellular structures formed by the fusion of multiple cells, facilitate intercellular cooperation and functional integration, enhancing cellular stability and efficiency. Inspired by the structure of syncytia and combining this with the structure of stirred reactors, we propose a novel type of cellular reactor. This design merges multiple units into a single entity, enhancing strength and centralizing functions.

Q12 What is the stability of the printed structures? Is etching and aging of the plastic a concern? Any characterization? Why is there a slight increasing trend as the cycle number increases for table S5? Is it perhaps the strut is being etched by the acid and thinning and allowing for more 1M H₂SO₄ to be loaded for the subsequent cycle therefore increasing the conversion rate?

Response: We appreciate your feedback. Following the completion of five cyclic experiments, the cellular reactor was cleaned, dried, and subsequently photographed, along with obtaining SEM images.

Figure S14 Photographs and SEM images of the cellular reactor before and after the cycling reaction (a-d) raw state, not processed in any way (e-h) after cyclic reactions

It is observable that after five cycles, the cellular reactor underwent minimal changes in appearance (Fig S14a, e). The overall structure remained intact (Fig S14b, f), with only slight fine lines appearing on the surface microstructure (Fig S14c, d, g, h). The degree of corrosion observed is entirely acceptable considering the number of cycles it underwent.

Even after five cycles of use, the cellular reactor remained in excellent condition, preserving its capacity to facilitate liquid-liquid two-phase interface formation. This led to a notable increase in the utilization efficiency of the cellular reactor and improved the economic feasibility of the reaction.

Q13 Bromothymol blue may be redissolving in the cyclohexane (no water transfer just bromothymol blue) and changing back to yellow.

Response: Thanks for your feedback. After careful consideration, we acknowledge that the description regarding the transfer and reaction of bromothymol blue was not sufficiently rigorous. During the diffusion process, once bromothymol blue transfers to the oil phase interface, there are indeed two possible outcomes: it can either cross the oil phase interface to reach the liquid-liquid interface or return to the bulk phase (oil phase). Therefore, after

transferring to the interface, bromothymol blue may not have reacted with sodium hydroxide but rather remained dissolved in cyclohexane. Even for bromothymol blue returning from the interface to the organic phase, there is still a chance of retransferring to the interface (as bromothymol blue is consumed at the interface, driven by chemical potential). Thus, for the interface reaction, this model remains conceptually correct and has been further refined. The relevant content is as follows:

To visualize interfacial mass transfer behavior, colored compounds were employed in the experiment. To capture color changes on the cellular reactor effectively, the cellular reactor remained stationary throughout the process. The color indicator, Bromothymol Blue (BTB), was dissolved in cyclohexane as the external phase, resulting in a transparent pale yellow solution (Fig. 4a). The internal phase of the cellular reactor consisted of an NaOH aqueous solution. Driven by concentration gradients, BTB molecules diffused from the bulk phase to the oil phase interface. If BTB molecules successfully reached the water-oil interface, they could react with NaOH, forming Bromothymol Blue Sodium (SBB) at the phase boundary. Conversely, BTB molecules would return to the oil phase. In the water/cyclohexane system, SBB readily dissolved in the aqueous phase, immediately imparting a blue color to the aqueous phase. With prolonged reaction time, the quantity of blue solution within the cellular reactor increased gradually. It is noteworthy that the blue liquid did not leak into the oil phase, and the color of the oil phase remained unchanged. This observation confirms that the reaction occurs exclusively at the interface, with oil-soluble reactants converting into water-soluble products and traversing the water-oil interface into the other phase. For further details, please refer to supporting Movie 1.

Q14 Specific concerns:

Pg 9, line 4: "...holding capacity when the characteristic size L surpasses h ." L is defined as the unit cell size—does this indicate that when the unit cell size surpasses h the cell cannot hold liquid regardless of the unit cell type? I would expect unit cell types with high relative density (large amount of internal struts) to be able to hold fluid even if the overall unit cell size is greater than h . I think the thought behind this sentence is correct, but specifying L is confusing.

Response: Thanks the reviewer for the valuable advice. We apologize for the confusion caused by the unclear description. Your inference is correct. What we intended to convey is that when the characteristic dimension L of a cell unit exceeds the characteristic height h , the cell cannot stably retain the liquid, regardless of the other size parameters of the cell.

Specifically, in the text, it is stated as follows: When the characteristic dimension L of a cell exceeds the height h , irrespective of the cell unit configuration, the honeycomb reactor loses its liquid retention capacity.

P9 Line 14:

When the characteristic size (L) of a cell exceeds h , the honeycomb reactor loses its liquid-holding capacity, regardless of the cell unit's configuration.

Q15 Specify which reactors are used in each reaction

Response: Thanks the reviewer for the valuable suggestions. In all the classical reactions conducted in this study, a 6-blade cellular reactor was employed, and the cell unit types have been explicitly identified in the manuscript. All the reactor information relevant to this study has been listed and labeled in Table S4. Therefore, simply referencing the reactor number is sufficient to ascertain the reactor details. We have incorporated information about the cellular reactor in all reaction descriptions to provide readers with a clear understanding of the experimental procedures.

Table S4 Parameters of cellular reactors in this study

Ractor	Structure	Location	L	D*	Blades	N
Ractor-A		Fig. 3e	2 mm	0.33	3	498
Ractor-B		Fig. 3e	2 mm	0.33	4	568
Ractor-C		Fig. 3e	2 mm	0.33	6	708
Ractor-D			2 mm	0.33	6	708
Ractor-E			2 mm	0.33	6	708
Ractor-F			-	-	6	-
Ractor-G			-	-	-	-
Ractor-H			-	-	-	-

Q16 Pg 16 line 18: “metal reactor”: isn’t the reactor plastic?

Response: Thanks for your feedback. The cellular reactor used in this study was manufactured using 3D printing with photosensitive resin. It's important to clarify that here, "reactor" does not refer to the cellular reactor itself but to a

vessel that houses both the cellular reactor and the external phase reactant. Industrial-scale reactors in typical production settings are usually metallic, and our intention was to confine corrosive components within the cellular reactor, preventing them from corroding the inner walls of the reactor vessel due to any potential detachment during rotation. To avoid any potential reader misunderstanding, the relevant content has been removed.

Q17 How does the contact angle change with temperature given the elevated temperature of some of your reactions

Response: Thanks the reviewer for the valuable comments. All our contact angle tests were conducted at 20°C. Our current contact angle measurement equipment is not capable of accurately measuring contact angles at elevated temperatures. Observations from simple temperature-varying experiments indicate that there is no significant change in the contact angle of the liquid during short-term heating to the reaction temperature. However, prolonged measurements could result in liquid evaporation, affecting measurement accuracy. Our stirrer remains operational at 60°C, the reaction temperature, without any observed leakage from the cellular reactor. Additionally, two other reactions we conducted, the coupling of thiol compounds and the oxidation of sulfides, were carried out at room temperature, where temperature had no discernible impact.

Q18 Fig 3d: typo cubic, n =3

Response: Thanks for your feedback. The relevant image content has been corrected. Additionally, we have included working points for the retention/leakage status of different cell unit types.

Table S11 Parameters of cellular reactors

Sample	Cellular type	L	D*	Liquid state
--------	---------------	---	----	--------------

S13-a	Cubic	4	0.2	Leaked
S13-b	Cubic	3	0.2	Leaked
S13-c	BCC	4	0.333	Held
S13-d	BCC	3	0.333	Held
S13-e	BCC	2	0.333	Held
S13-f	BCC+frame	4	0.333	Held
S13-g	BCC+frame	3	0.333	Held
S13-h	BCC+frame	2	0.333	Held

Figure S13 Liquid state demonstration of different cellular reactors

Figure S13 demonstrates the liquid retention effect. For cells S13-a&b, since they do not satisfy the formula $(\pi \frac{\lambda}{n} \cos \theta + 1)^{-1} < D^* < 1$, their operating points in Figure 3d, at $n=2,3d$, are located above the $y=x$ line, and for $n=1$, very close to the $y=x$ line, resulting in liquid leakage within the cells. However, for cells S13-c-h, the operating points are below the $y=x$ line, thereby effectively retaining the liquid. In this case, the liquid is water, dyed with methylene blue.

Furthermore, the operating points of the aforementioned cells have been incorporated into the new Figure 3d. In this figure, asterisks denote the working points used throughout the study, circles represent liquid retention, and crosses indicate liquid leakage.

the lower limit of the range of values for the equivalent diameter, D_{min}^* , as a function of D^*

Q19 Fig 3b and 3f: keep color scheme consistent

Response: Thanks the reviewer for the valuable suggestions. We have harmonized the color schemes of Figures 3b and 3f to match the consistency with other figures. However, for visual harmony, we have made appropriate adjustments to the opacity.

Q20 I do not see any captions or descriptions for videos! Overall, I think the operation of this device is very exciting, but the videos don't capture the beauty

of this application. Was the colorimetric reaction done statically or with rotation?

Response: Thanks the reviewer for the valuable suggestions. We have incorporated textual annotations into the video to provide viewers with additional information. We have also clarified in the text that the colorimetric reaction was conducted under static conditions. The slight oscillation observed in the video frames was a result of the screw tightening process.

Neutral red is a commonly used dye, typically employed as an acid-base indicator. Its chemical formula is $C_{15}H_{16}N_3 \cdot HCl$, existing in the form of a hydrochloride salt. Neutral red exhibits a color change range between pH 6.4 and 8, transitioning from red to yellow.

Under initial conditions, neutral red dissolves in ethyl acetate, appearing pale red. A 1M NaOH solution, serving as the aqueous phase, is adsorbed into the cellular reactor. The stirrer is inserted into the organic phase and immobilized, subsequently rotating at 50 rpm, as detailed in video Movie 5. With the progression of stirring, the pale pink solution gradually transforms into pink-orange, ultimately turning into pale yellow. This indicates that neutral red molecules at the interface undergo a chemical reaction with NaOH, where the HCl moiety is neutralized by NaOH and successfully returned to the organic phase, resulting in a change in color.

Q21 Please consider reviewing and citing : <https://doi.org/10.1038/s41567-021-01204-4>

Response: Thanks the reviewer for the valuable suggestions. We have indeed read the literature you mentioned. It has provided us with valuable insights and served as a source of inspiration for addressing many issues during the revision process. We have also cited this reference in our work. The positions of the aforementioned reference correspond to citations 23.

Reviewer: 4

Overall Comments:

This paper presents a biomimetic 3D printed cellular reactor for liquid-liquid reactions, offering intriguing results that, with some minor revisions, I believe can be suitable for publication. Below are specific comments and suggestions for improvement:

Response: We appreciate the reviewer's highly positive comments. We tried our best to address all the detailed comments and made a detailed revision of our manuscript.

Q1 1. On page 15, line 8, the authors posit that the product is primarily soluble in the oil phase. However, in Figure 4a, a water-soluble product is shown transferring to water. It would be beneficial if an experiment illustrating the transfer of the product to the oil phase could be included.

Response: Thanks the reviewer for the valuable suggestions. We have incorporated textual annotations into the video to provide viewers with additional information. The text also clarified that the colorimetric reaction was conducted under static conditions. The slight oscillation observed in the video frames resulted from the screw-tightening process.

Neutral red is a commonly used dye, typically employed as an acid-base indicator. Its chemical formula is $C_{15}H_{16}N_3 \cdot HCl$, existing in the form of a hydrochloride salt. Neutral red exhibits a color change range between pH 6.4 and 8, transitioning from red to yellow.

Under initial conditions, neutral red dissolves in ethyl acetate, appearing pale red. A 1M NaOH solution, serving as the aqueous phase, is adsorbed into the cellular reactor. The stirrer is inserted into the organic phase and immobilized, subsequently rotating at 50 rpm, as detailed in video Movie 5. With the progression of stirring, the pale pink solution gradually transforms into pink-orange, ultimately turning into pale yellow. This indicates that neutral red molecules at the interface undergo a chemical reaction with NaOH, where NaOH neutralizes the HCl moiety and successfully returns to the organic phase, resulting in a color change.

Q2. In the Introduction section, it is advisable for the authors to include references to application-specific literature to clarify the engineering significance of their results.

Response: We appreciate your feedback, which is of utmost importance to us. We have realized that dedicating substantial space to describe the behavior inspired by our initial insights is inappropriate when outlining the fundamental concept of the liquid-liquid reactor. Therefore, we have reduced the description of biological membrane behavior and coacervate formation, opting for a brief mention. Our focus is now primarily on the liquid-liquid reactor. The revised background section is as follows:

Liquid-liquid reactions are vital in chemical synthesis due to the fast mass transfer between interfaces, uniform temperature field distribution, dynamic interfacial composition updating, and resistance to catalyst poisoning. Typically, one reactant acts as a dispersed phase, forming tiny droplets in the continuous phase of another substance, increasing the area for molecular diffusion. The surface tension causes the dispersed-phase droplets to adopt a spherical shape, minimizing the system's energy. However, collisions between droplets lead to reduced contact between the liquid phases, often necessitating fast mechanical stirring or emulsifiers.

Stirred reactors are widely used in chemical engineering and industrial processes for liquid-liquid reactions. Stirring facilitates thorough mixing of the two phases, enhancing reaction rates. It also helps maintain temperature uniformity, preventing temperature gradients due to exothermic or endothermic reactions. However, stirred reactors have drawbacks, such as high shear forces that can damage shear-sensitive chemicals, and insufficient stirring speeds that fail to maintain adequate interfacial area between the two liquid phases, leading to reduced reaction rates.

Using emulsifiers stabilizes droplets and reduces their size, thereby increasing the total interfacial area. However, the compatibility of emulsifiers with the

reaction system must be considered, as the introduction of inappropriate emulsifiers can decrease reaction efficiency. Additionally, the resulting emulsion complicates the separation and purification of products post-reaction. Most importantly, the use of emulsifiers inevitably requires rapid mechanical stirring.

Phase transfer catalysts can enhance the efficiency of liquid-liquid biphasic reactions without requiring vigorous stirring. However, the subsequent separation process becomes more complex due to the high affinity of phase transfer catalysts for both phases. Therefore, designing a novel reactor that can enhance liquid-liquid interfacial mass transfer at low stirring speeds and allow for easy separation after the reaction is crucial.

In nature, numerous liquid-liquid reactions exhibit remarkable controllability. Cell membranes represent a widely occurring natural interface for such reactions, where reactants diffuse to the interface, and products depart, maintaining interface stability throughout the process. Syncytia, large cellular structures formed by the fusion of multiple cells, facilitate intercellular cooperation and functional integration, enhancing cellular stability and efficiency. Inspired by the structure of syncytia and combining this with the structure of stirred reactors, we propose a novel type of cellular reactor. This design merges multiple units into a single entity, enhancing strength and centralizing functions.

Q3. Could the authors provide a description of the separation process that follows the reaction and elaborate on the typical methods used for product separation?

Response: Thanks the reviewer for the valuable suggestions. After the completion of the reaction, the cellular reactor is lifted upwards from the reaction solution until it is outside the reactor vessel. Since the internal phase liquid is confined within the cellular reactor, it separates from the reaction solution as the cellular reactor is detached. You can refer to the separation

section in the latter part of Movie 5 for the detailed process. In the reaction solution, the substrate conversion rates are consistently above 95%, with selectivity approaching nearly 100%. Therefore, solvent removal by rotary evaporation allows for the recovery of the crude product.

Q4. The authors are advised to add specific findings to the Conclusion section.

Response: Thanks the reviewer for the valuable suggestions. We have incorporated more specific research findings into the conclusion section. The relevant content is as follows:

Inspired by the syncytium structure in organisms and the construction of biological membranes, we have developed a cellular reactor for liquid-liquid reaction systems. This reactor has dual functions of enhancing interphase contact and restricting the spatial distribution of reaction components, achieving high synthesis yields in liquid-liquid reactions, and improving the safety and separability of reactions. It allows for precise control over the spatial distribution of single-phase components in liquid-liquid reactions, ensuring catalytic efficiency. The reactor applies to completely immiscible and partially miscible water-oil biphasic systems, capable of performing acid-catalyzed addition reactions, base-catalyzed condensation reactions, and oxidation reactions involving strong oxidants. The majority of reactions can achieve conversion rates higher than 95% within a short period. Chemical reactions in this cellular reactor are characterized by their simplicity and efficiency, straightforward separation of catalyst components and products, enhanced safety, good reproducibility, and reusability. It provides a new venue and design paradigm for liquid-liquid reaction systems.

Q5. Have the MD simulations been conducted using the VASP software? For the MD simulations, it is recommended that the authors include details about the algorithms used for temperature and pressure control, information about

how the system reached equilibrium, and the time step employed.

Response: Thanks the reviewer for the valuable suggestions. The MD simulations were conducted using the VASP software, with a system containing 18,000 atoms and dimensions of 16×12×1 nm, over 100,000 steps of 100 fs each.

Temperature control within the system was achieved using the Nose-Hoover thermostat algorithm. This method is designed to maintain temperature stability within an NVT (constant volume and temperature) ensemble by simulating the thermodynamic behavior of the system's temperature. The Nose-Hoover algorithm introduces an additional dynamical variable (the Nose-Hoover variable), which is coupled with the system's temperature. The dynamics of this variable, akin to a thermal bath in the canonical ensemble, aim to adjust the system's temperature. In addition to the motion equations of particles, the Nose-Hoover algorithm incorporates an equation for the motion of the Nose-Hoover variable itself. This supplementary equation manages temperature control and ensures energy exchange between the system and the thermal bath. The dynamical coupling of the Nose-Hoover variable with the system's kinetic energy influences the temperature evolution, stabilizing the system's temperature through adjustments to the Nose-Hoover variable.

The system pressure was controlled using the Berendsen pressure algorithm, a commonly utilized pressure control method in molecular dynamics simulations aimed at quickly achieving isothermal compression or expansion. The core principle involves the introduction of a pressure coupling term into the dynamical equations, linking the system's volume with the desired pressure to adjust the system's pressure. A relaxation time (often denoted as TAUP) is specified within the Berendsen algorithm to control the rate of relaxation under pressure coupling, with shorter relaxation times leading to faster pressure relaxation and longer times resulting in slower relaxation. The fundamental concept of pressure control is to couple the system's volume with pressure, allowing for rapid attainment of the desired pressure during simulation.

Equilibration process for bringing the simulation system to equilibrium:

Initial Structure Setup: Prior to initiating molecular dynamics simulations, initial atomic coordinates must be provided.

Preheating Phase: Before the simulation starts, a thermostating phase is conducted to gradually bring the system to an equilibrium state through the use of thermostat algorithms. Temperature and pressure control are achieved by setting appropriate parameters and selecting the Nose-Hoover temperature control algorithm and the Berendsen pressure algorithm, ensuring the system progressively reaches the desired temperature during the preheating phase.

Equilibrium Check: Following the preheating phase, the system should undergo an equilibrium check, assessing whether equilibrium has been reached by monitoring changes in physical properties and the stability of pressure.

Production Simulation: Once the system has reached equilibrium, the physical properties of interest can be recorded and analyzed.

This article primarily focuses on describing how to construct a cellular reactor and the proposal and validation of its functions. The content pertaining to molecular dynamics serves as a means to verify reactions at the liquid-liquid interface and has a slightly weaker relevance to the overall study. Therefore, we have moved this portion of the research details to the supplementary information. This ensures the accuracy and relevance of the main text's research while allowing readers interested in this aspect to engage with it selectively in the supplementary information.

Reviewer #5 (Remarks to the Author):

This was a joint review conducted with one of the other reviewers; all comments are already contained in the other review.

Reviewer #6 (Remarks to the Author):

This was a joint review conducted with one of the other reviewers; all comments are already contained in the other review.

REVIEWER COMMENTS

Reviewer #1 (Remarks to the Author):

Most of the issues have been addressed nicely, during revising the manuscript. And the manuscript is now much better and suitable to be published.

Reviewer #2 (Remarks to the Author):

Can and Tao undoubtedly put impressive efforts in improving their paper after revision. I particularly appreciated the newly added particle image velocimetry experiments, and SEMs of the reactor materials before and after treatment with corrosive liquids. NMR of the obtained products is a welcome addition as well. Also, the control experiments performed with addition of glass bubbles (Table S8) are very nice and smart.

Although the authors clearly did a massive amount of additional work, I am still not completely convinced that the cellular reactor represent an advancement in the field. The cellular reactor obviously makes phase separation trivial, which is a huge benefit for liquid-liquid reactions, however the little increase in the interface area (2.3 times) in conjunction with the limited stirring speed (lower than 200 rpm) represent important limitations when coming to reaction conversion rates. In fact, the results listed in table S18 show that the reaction performed at 200 rpm (when liquid leakage occurs) is clearly faster than all the others. Therefore, the benefit of extremely easy liquid-liquid phase separation is obtained at the expense of the reaction rates. I can see the advantage of the method, but I am not completely convinced that it counterbalances the drawback. I strongly encourage the authors to comment on this aspect in the dedicated section of the SI, but I do not oppose to publication.

Apart from this, I have just a couple of comments more to add:

-in response to Q3: TOC and TGA are appropriate for the analysis of ethyl acetate in water and the reaction residues respectively, even though the amount of carbon-based material in sample S15-3W (Table S15) is bizarre. I would have used Karl Fisher method for the determination of water in the ethyl acetate, instead of NMR.

-In Table S7, addition of the conversion rate obtained with reactor C after 60 minutes would allow for a more straightforward comparison of results with the cellular reactor.

Reviewer #3 (Remarks to the Author):

Overall Comments:

Thank you for addressing the majority of the concerns raised. Overall, I believe the bulk of my concerns have been addressed. However, the organization makes the paper hard to follow and difficult to validate. Major content concerns are Q5 and Q10b; this may be remedied by a simple clarification.

1. I appreciate the added information provided in the SI. However it has become unwieldy. I would suggest headings to separate subsections. I also would err on the side of repeating information, rather than forcing readers to search through the document for relevant information, or at least clearly referencing relevant sections. The references in the main manuscript don't seem to follow the order in the SI. I.e reference Fig S1, S2, S3 etc.
2. I appreciate the effort put into revising the introduction and tailoring the manuscript to address the relevance of the current design with respect to existing reactors. The response received for Q1 addresses concerns regarding comparison to existing work. However, I do not see these comparisons clearly listed in the manuscript or SI. Especially the quantitative comparison of catalytical efficiency seems like its worthwhile to include in one of your charts in the SI. The assessment of the reactor performance is dispersed throughout the SI, making it difficult to assess the reactor efficiency with respect to controls. See Q14
3. The analysis of the surface tensions/interfacial tension etc has improved (SI pg 11-14), however it is still very difficult to follow. Suggestions/Concerns:
 - a. Equation following equation 6 add " Use equation 5 and 6 to solve for σ_s^d and σ_s^p to calculate σ_s "
 - b. To calculate σ_{slw} does not appear to match up with $\sigma_s = 62.46$, $\sigma_{lw} = 72.8$ with $\Theta_w = 33.2^\circ$; expect $\sigma_{slw} = 1.54$ mN/m
 - c. Table S2 and S3: add row with actual values
 - d. Unclear what solid this analysis is for? Clear resin that has been o2 plasma treated?
 - e. Table S3 Unclear what the measured contact angles are (especially θ_D)
 - f. Table S2: missing σ_{lc}
4. Pg 12 eq 11: relating equations 8-10
5. Equivalent wetting parameter: I do not believe this parameter is adequately described in the main text. The original concerns I had with this parameter were not addressed. (originally Q5) I think this is probably one of the most significant findings of this paper- a way to determine the behavior of arbitrary designs without experimental testing, yet the description and analysis is insufficient to validate. Page 13 line 2 indicates "the equivalent wetting coefficient is derived from measurements .." however there is no discussion of how to measure it. Is it actually just derived from the calculation of the A_{SL1} interface and eq 10? See Q10b.
 - a. How are Eq 9 and 10 derived?
 - b. Is Eq 9 valid for all unit cells? There does not appear to be any dependance on the unit cell type, yet Fig 2b shows different A_{LL} interfaces dependent on the unit cell type
6. Should "Ractor" be "reactor" throughout manuscript
7. Figure s6: what is the difference between a1 vs a2 etc
8. PG 4 LINE 14: "capability to flow liquid inside the reactor renewing reaction interface": this seems inaccurate or a bit misleading. The liquid within the cellular fluidic reactor is not flowing although the individual cells are connect and can replenish by passive diffusion etc.

9. Pg 5 line 7: stirring paddle “ eliminating external diffusion” this is a bit confusing not sure what is meant by this
10. Figure 2:
 - a. What is L? is this the size of the unit cell? Put units
 - b. How is figure 2b calculated? Was computational software used to extract the ALL and ASL interface? What assumptions are made?
 - c. How is ASL* and ALL* normalized?
11. Pg7 line 21: “refer to appendix” where in the appendix?
12. Fig 3:
 - a. 3b/3d/3f color scheme not consistent n=3 red/orange and n=1 green for both—The original comment was to have the different colors indicate the coordination number across fig 3b, d and f ie n = 1, 2, 3. In fig 3b n =3 is red, while in fig 3d n =1 is blue and in fig 3f n = 3 is green. (I have no problem with purple/other colors; I just think its much easier to follow if the colors indicate the coordination number.)
 - b. 3d what do the symbols mean? This is not in the main text/figure legend: **asterisks denote the working points used throughout the study, circles represent liquid retention, and crosses indicate liquid leakage.**
13. Table S4: put in more information about the reactors—ie the resin types for reactors C-E
14. Table S7: combine with outputs from cellular reactor; hard to compare! Consider combining tables S10/S7, possibly S4? Or putting these results in closer proximity to each other?
15. Table S12: think the caption should say for entry 21. Seems like inconsistent notation, unless this is describing different reactors? Why is “entry” being used here? Is SA-4 “Ractor-A” in figure S4?

Reviewer #4 (Remarks to the Author):

The authors have responded to my previous comments and revised the manuscript accordingly. I believe it is acceptable for publication in its form.

Reviewer #5 (Remarks to the Author):

This was a joint review conducted with one of the other reviewers; all comments are already contained in the other review.

Reviewer #6 (Remarks to the Author):

This was a joint review conducted with one of the other reviewers; all comments are already contained in the other review.

RESPONSE TO REVIEWERS' COMMENTS

Comments from the Reviewer:

Reviewer: 2

Overall Comments:

Can and Tao undoubtedly put impressive efforts in improving their paper after revision.

I particularly appreciated the newly added particle image velocimetry experiments, and SEMs of the reactor materials before and after treatment with corrosive liquids. NMR of the obtained products is a welcome addition as well. Also, the control experiments performed with addition of glass bubbles (Table S8) are very nice and smart.

Although the authors clearly did a massive amount of additional work, I am still not completely convinced that the cellular reactor represent an advancement in the field.

The cellular reactor obviously makes phase separation trivial, which is a huge benefit for liquid-liquid reactions, however the little increase in the interface area (2.3 times) in conjunction with the limited stirring speed (lower than 200 rpm) represent important limitations when coming to reaction conversion rates. In fact, the results listed in table S18 show that the reaction performed at 200 rpm (when liquid leakage occurs) is clearly faster than all the others. Therefore, the benefit of extremely easy liquid-liquid phase separation is obtained at the expense of the reaction rates. I can see the advantage of the method, but I am not completely convinced that it counterbalances

the drawback. I strongly encourage the authors to comment on this aspect in the dedicated section of the SI, but I do not oppose to publication.

Response: We appreciate the reviewer's highly positive comments. We tried our best to address all the detailed comments and made a detailed revision of our manuscript. Regarding the issues raised by the reviewers, we have provided supplementary information in the SI (Line 10, Page 70, SI), as follows:

Although the capability of cellular reactors to enhance the rate of chemical reactions under working conditions is limited, their functionality in maintaining interface stability, limiting corrosive components, and greatly reducing separation difficulty is irreplaceable. Therefore, the introduction of cellular reactors serves a meaningful purpose.

Apart from this, I have just a couple of comments more to add:

Q1: in response to Q3: TOC and TGA are appropriate for the analysis of ethyl acetate in water and the reaction residues respectively, even though the amount of carbon-based material in sample S15-3W (Table S15) is bizarre. I would have used Karl Fisher method for the determination of water in the ethyl acetate, instead of NMR.

Response: We appreciate the valuable feedback from the reviewer. We have re-measured the water content in ethyl acetate using the Karl Fischer method. The results indicate a lower water content, suggesting a certain level of error in the NMR determination. This information has been added to the "Cross-contamination between

water phase and organic phase" section in the SI. Additionally, the content in the SI has been reorganized, resulting in changes to the sequence of figures and tables. The details are as follows:

The Karl Fischer method was employed to measure the water content in ethyl acetate. Given that the water content in the sample is not trace but rather minute, volumetric analysis is deemed suitable for this system. A two-component Karl Fischer reagent (comprising solvent and titrant) was utilized, with pre-titration of the solvent conducted prior to experimentation to establish a blank control for moisture content. A specific sample volume was introduced into the solvent, followed by incremental titrant addition with stirring to ensure a complete reaction. Upon color change indicating the endpoint of titration, the volume difference of titrant before and after titration (ΔV) was recorded and used to calculate the water content. Since commercially available ethyl acetate may also contain water, immediate water content testing was performed upon extraction from the reagent bottle as a control group.

The water content is calculated using the following formula.

$$C_W = \frac{\Delta V \times Tit}{V_{EA}}$$

In this equation, ΔV represents the volume difference of the titrant before and after titration, V_{EA} denotes the volume of the sample added (1 mL), Tit signifies the titer (5000 ppm, expressed as H₂O), and C_W denotes the water content of the sample.

Table S17 Water content of ethyl acetate determined by Karl Fischer volumetric titration

Sample	EtOAc	S16-1EA	S16-2 EA	S16-3 EA
ΔV (mL)	0.08	0.17	0.24	0.35
C_W (ppm)	400	850	1200	1750

At room temperature, the utilization of cellular reactors resulted in an increase in water content, from 400 ppm to 850 ppm, which remained below that observed with commercially available stirrers (1200 ppm) and notably lower than in cases of vigorous agitation (1750 ppm). It indicates that while the cellular reactor increases the interfacial area, it does not introduce additional cross-contamination between the two phases. On the contrary, compared to vigorous stirring, the cellular reactor reduces cross-contamination between the phases and facilitates rapid separation afterward, making it an advanced tool for chemical processes.

Q2: In Table S7, addition of the conversion rate obtained with reactor C after 60 minutes would allow for a more straightforward comparison of results with the cellular reactor.

Response: We appreciate the reviewer's feedback. We have incorporated the catalytic data for Reactor-C into the old Table S7 (now renumbered as Table 11), facilitating a more intuitive comparison regarding the necessity of using cellular reactors. The relevant content is provided below.

Table S11 Comparison of the solid cellular reactor with two types of commercially available reactors

Time (min)	Conversion rate (%)			
	Reactor-C	Reactor-F	Reactor-G	Reactor-H
30	37.6	1.9	4.1	3.8
60	64.8	2.5	6.9	8.1
90	84.2	4.9	8.6	10.2

Reviewer: 3

Overall Comments:

Thank you for addressing the majority of the concerns raised. Overall, I believe the bulk of my concerns have been addressed. However, the organization makes the paper hard to follow and difficult to validate. Major content concerns are Q5 and Q10b; this may be remedied by a simple clarification.

Response: We appreciate the reviewer's highly positive comments. We tried our best to address all the detailed comments and made a detailed revision of our manuscript.

Q1 I appreciate the added information provided in the SI. However it has become unwieldy. I would suggest headings to separate subsections. I also would err on the side of repeating information, rather than forcing readers to search through the document for relevant information, or at least clearly referencing relevant sections. The references in the main manuscript don't seem to follow the order in the SI. Ie reference Fig S1, S2, S3 etc.

Response: Thank you for the reviewer's suggestions. We have revised the sequence

of references in the main text and reorganized the figures and tables in the SI accordingly, aligning them with the revised order. Similar-themed figures and tables are now grouped within the same sections, with added section titles to differentiate between the functionalities of each SI part. Additionally, we have compiled the new and old numbering of figures and tables in the SI into a table for improved review efficiency.

Comparison of the new and old numbering of figures in SI.

New	S5	S6	S7	S8	S9	S10	S11	S12
Old	S13	S44	S45	S5	S15	S16	S17	S18
New	S13	S14	S15	S16	S17	S18	S19	S20
Old	S19	S20	S21	S22	S23	S24	S25	S26
New	S21	S22	S23	S24	S25	S26	S27	S28
Old	S27	S28	S29	S30	S31	S32	S33	S34
New	S29	S30	S31	S32	S33	S34	S35	S36
Old	S35	S36	S37	S38	S39	S14	S8	S9
New	S37	S38	S39	S40	S41	S42		
Old	S10	S11	S12	S6	S7	S43		

Comparison of the new and old numbering of tables in SI.

New	S5	S6	S7	S8	S11	S12
Old	S11	S12	S13	S14	S7	S5
New	S13	S14	S15	S16	S17	S18
Old	S6	S18	S8	S15	S16	S17

Q2 I appreciate the effort put into revising the introduction and tailoring the manuscript to address the relevance of the current design with respect to existing reactors. The

response received for Q1 addresses concerns regarding comparison to existing work. However, I do not see these comparisons clearly listed in the manuscript or SI. Especially the quantitative comparison of catalytical efficiency seems like its worthwhile to include in one of your charts in the SI. The assessment of the reactor performance is dispersed throughout the SI, making it difficult to assess the reactor efficiency with respect to controls. See Q14

Response: Thank you for the reviewer's suggestions. Regarding the catalytic and extraction functionalities of cellular reactors, we have placed the comparative information with literature in the "Comparison of the catalytical efficiency by use and non-use of cellular reactor" section of the SI. The relevant content is provided below:

Table S19 Catalytic efficiency between cellular reactor and pickering emulsions

	2H-pyran (mmol)	Time	Yield	Catalyst	Temp.
This work	12.5	3 h	97%	1 M H ₂ SO ₄	60°C
Reference S1	75	500h	81%	1 M H ₂ SO ₄	50°C
Reference S2	0.24	18 h	81%	Trialkylphosphonium oxoborate	r.t.
Reference S3	10	1.5	95%	Thiourea, tetrafluoroborate (1:1)	r.t.
Reference S4	5.5	7	98%	Ti ⁴⁺ /4Å	40°C
Reference S5	2	24	88%	Pyridinium	r.t.

Compared to Pickering emulsion¹, the introduction of cellular reactors facilitates rapid reaction occurrence, converting more substrate dihydrofuran at a higher rate per unit time. Unlike Pickering emulsion, which requires rapid agitation for emulsification and subsequent demulsification post-reaction, the introduction of cellular reactors streamlines the operation. Direct use of sulfuric acid as the catalytic component eliminates the synthesis of catalysts, as well as the post-reaction separation process. Therefore, utilizing cellular reactors for chemical reactions offers unique advantages.

A comparison of extraction performance between cellular reactors and advanced extraction tools. We employed a cellular reactor for liquid-liquid extraction processes. Riboflavin, also known as Vitamin B2, is an essential nutrient for the human body. Initially, riboflavin dissolves in ethyl acetate, imparting a pale yellow color. A 1M NaOH aqueous phase is adsorbed into the cellular reactor. A stirring paddle is inserted into the organic phase and fixed, followed by rotation at a speed of 50 rpm. As stirring progresses, the pale yellow color in the solution gradually lightens, indicating the transfer of riboflavin molecules from the organic phase to the aqueous phase, completing the extraction process.

Using UV-Visible spectroscopy, we measured the riboflavin concentration in the organic phase before and after extraction. The extraction efficiency was calculated as 58.9% (concentration-based) based on the following formula.

$$\eta = \frac{C_0 - C_1}{C_0} \times 100\%$$

In this context, η represents the extraction efficiency, where C_0 denotes the initial concentration and C_1 represents the concentration in the residual phase after

extraction.

Compared with state-of-the-art extraction tools, we also reviewed the literature on riboflavin extraction. Riboflavin was successfully extracted using lactic acid ethyl ester and organic salt-based ATPS in the system of lactic acid ethyl ester (1) + sodium citrate (2) + water (3), achieving a maximum efficiency of 87.6%. While the extraction efficiency of the cellular reactor falls short of previously reported extraction tools, we believe that through modification and optimization, there is potential for expanded applications and improved efficiency in the future.

Q3 The analysis of the surface tensions/interfacial tension etc has improved (SI pg 11-14), however it is still very difficult to follow. Suggestions/Concerns:

- a. Equation following equation 6 add " Use equation 5 and 6 to solve for σ_s^d and σ_s^p to calculate σ_s "
- b. To calculate σ_{sIW} does not appear to match up with $\sigma_s=62.46$, $\sigma_{IW}=72.8$ with $\Theta W=33.2^\circ$; expect $\sigma_{sIW} = 1.54$ mN/m
- c. Table S2 and S3: add row with actual values
- d. Unclear what solid this analysis is for? Clear resin that has been O_2 plasma treated?
- e. Table S3 Unclear what the measured contact angles are (especially θD)
- f. Table S2: missing σ_{lc}

Response: We appreciate the valuable feedback from the reviewer. The instruction " Use equation 5 and 6 to solve for σ_s^d and σ_s^p to calculate σ_s ." has been added to the SI (Line 13, Page 12, SI). This analysis applies to solids treated with oxygen plasma. The cellular reactor truss processed by O_2 plasma has also been included in the SI (Line 6, Page 11, SI). The value of σ_{sIW} is calculated based on the measured surface tension of water, which is actually 1.54 mN/m (Line 2, Page 13, SI).

Additionally, the θ value for the three-phase contact angle should be 38.7° (Line 14, Page 13, SI). The surface tension data (σ_{LC}) for cyclohexane has been added to Table S2, and the actual values of surface tensions and contact angles have been incorporated into the original Tables S2 and S3.

Table S2 Description of interfacial tension

Interfacial tension	σ_{IW}	σ_{ID}	σ_s	σ_{LC}
Description	gas-liquid	gas-liquid	gas-solid	gas-liquid
Value (mN/m)	72.8	50.8	62.46	26.33
Source	measured	measured	calculated	measured
Interfacial tension	σ_{IWL}	σ_{SL}	σ_{SLW}	
Description	liquid-liquid	solid-liquid	solid-liquid	
Value (mN/m)	44.35	36.13	1.54	
Source	measured	calculated	calculated	

Table S3 Description of inter contact angle

Contact angle	θ	$\cos \theta_W$	$\cos \theta_C$	$\cos \theta_D$
Description	liquid-liquid-solid	gas-liquid-solid	gas-liquid-solid	gas-liquid-solid
Value ($^\circ$)	38.7	33.2	0	136.2
Source	calculated	measured	measured	measured

Q4. Pg 12 eq 11: relating equations 8-10

Response: Thank you for the reviewer's reminder. The relevant content has been corrected in manuscript.

To establish the range of characteristic cell diameters, equations 8 to 10 were

simultaneously solved, yielding the following relationship (Line 1, Page 13)

Q5. Equivalent wetting parameter: I do not believe this parameter is adequately described in the main text. The original concerns I had with this parameter were not addressed. (originally Q5) I think this is probably one of the most significant findings of this paper- a way to determine the behavior of arbitrary designs without experimental testing, yet the description and analysis is insufficient to validate. Page 13 line 2 indicates “the equivalent wetting coefficient is derived from measurements ..” however there is no discussion of how to measure it. Is it actually just derived from the calculation of the *ASL1* interface and eq 10? See Q10b.

a. How are Eq 9 and 10 derived?

b. Is Eq 9 valid for all unit cells? There does not appear to be any dependence on the unit cell type, yet Fig 2b shows different *ALL* interfaces dependent on the unit cell type

Response: Thank you for the reviewer's reminder. Formulas 9 and 10 are derived from analyzing the geometric shape of the cell unit. The specific process is as follows: for a given cell with a cubic periodicity (referring to any cell with a cubic framework, not specifically to SC cells, all cells mentioned in this paper have a cubic repetition period). Due to the symmetry of the cube, it is sufficient to analyze one face and then multiply by 6. In order to determine the feasibility of design behavior without conducting experiments, Formulas 9 and 10 are based on the assumption that the boundary of the internal liquid is located at the highest point of the truss. This assumption seems oversimplified, as liquids exhibit surface tension (especially water), and the geometric

shape of the liquid surface is not so regular. However, when placed in another liquid, the liquid-liquid interface tends to flatten out due to the other liquid also trying to wet the frame and balance the pressure on both sides of the liquid-liquid interface, typically resulting in a nearly planar liquid-liquid interface under static conditions. Additionally, for different types of cell units, since the basic repetition period is a cube, the analysis of A_{LL} is similar, only needing to replace the four edges of the cross-section with four vertices to analyze BCC cell units, and combining them can analyze BCC+frame units. SC and BCC+frame units have the same formula structure due to their frame structure along the edges of the cube (Formula 9a). BCC, on the other hand, presents a distinct formula structure due to the frame along the body diagonal (Formula 9b). Multiplying by 6 for a single cross-section (specifically, the degree of freedom n , which is 6 for a single cell) yields the overall A_{LL} . (Line 14-15, Page 12)

$$A_{LL} = nL^2(1 - D^*)^2 \quad (9a)$$

$$A_{LL} = nL^2\left[1 - \frac{\pi}{4}(D^*)^2\right] \quad (9b)$$

Unlike the liquid-liquid interface (A_{LL}), the solid-liquid interface A_{SL1} is much more complex because the liquid L1 is located inside the solid S, so their contact mainly occurs through the internal framework. Under the assumption of a cubic framework, evaluating BCC and BCC+frame cell units is more challenging than simple cubic cells because the body center is the focus of four diagonals, which increases the analytical difficulty. Therefore, to quantify the wetting situation of irregular cell units (compared to simple cubic ones) similar to the analysis of A_{LL} , an equivalent wetting coefficient λ is proposed. As a dimensionless number, λ simplifies the degree of infiltration of

the cell's internal structure and can be considered to quantify the wetting degree of solid S by liquid L1 based on edges to some extent.

The equivalent wetting coefficient λ is calculated through quality analysis of cell units using 3D modeling software and is implicitly related to the characteristic diameter D^* (Fig. 3c). (Line 6, Page 13)

Q6. Should "Ractor" be "reactor" throughout manuscript

Response: Thank you for the reviewer's attention to detail. I believe the reviewer may be referring to Table S4 in the SI, where the reactor designation was mistakenly spelled as "Ractor" instead of "Reactor". We have rectified this error and thoroughly reviewed the manuscript and SI for spelling consistency.

Q7. Figure s6: what is the difference between a1 vs a2 etc

Response: Thank you for the reviewer's inquiry. We believe the reviewer's question pertains to the section on PIV in the SI. The difference between the original figures S7 a1 and a2 lies in the presence or absence of a scale. The scale provided by the image analysis system occupies the lower part of the image (which cannot be altered). To ensure that interested readers have access to all information within the field of view, we have presented both versions of the images side by side. Alternatively, we could have merged the scale-carrying screenshot below the original image without a scale. However, to avoid any suspicion of withholding information, we opted for this combined image format.

Q8. PG 4 LINE 14: “capability to flow liquid inside the reactor renewing reaction interface”: this seems inaccurate or a bit misleading. The liquid within the cellular fluidic reactor is not flowing although the individual cells are connect and can replenish by passive diffusion etc.

Response: Thank you for the reviewer's suggestions. To avoid any misunderstandings, we have amended the relevant text as follows:

the internal connectivity of cellular reactors facilitates passive diffusion of liquids within, affording opportunities for interface renewal. (Line 14, Page 4)

Q9. Pg 5 line 7: stirring paddle “ eliminating external diffusion” this is a bit confusing not sure what is meant by this

Response: Thank you for the reviewer's inquiry. Throughout the manuscript, we have employed the term "cellular reactor." The sudden appearance of the term "stirrer" indeed seems abrupt. It is worth noting that the parent structure of the cellular reactor exhibits a stirrer-like configuration, as evidenced by the model (Figure 3e) and photographs (Table S4). Our intention was to convey that the cellular reactor possesses a stirrer-like structure and functionality. One of the functions of the stirrer is to mix liquids, reducing or even eliminating concentration gradients. As chemical reactions progress, there remains a difference between the substrate concentration in the liquid and that on the surface of the stirrer. Rotating at a certain speed can effectively eliminate outward diffusion. This point is also reflected in the SI, in the

original Table S14, now Table S34, where a significant increase in reaction rate is observed with a given rotation speed, demonstrating the substantial reduction of outward diffusion. Of course, for any inappropriate wording, we have made corrections in the main text. The revised text is provided below:

Upon rotation, the stirred structured cellular reactor further enhances mass transfer, simultaneously eliminating external diffusion and maintaining interfacial stability. (Line 6, Page 5)

Q10. Figure 2:

- a. What is L ? is this the size of the unit cell? Put units
- b. How is figure 2b calculated? Was computational software used to extract the ALL and ASL interface? What assumptions are made?
- c. How is ASL^* and ALL^* normalized?

Response: Thank you for the valuable feedback from the reviewer. " L " represents the length of the cell unit, which in this paper is 2 mm, as mentioned in the Methods section under "Fabrication of cellular reactors." The presentation of this information may not align well with typical reading conventions. Therefore, we have added a note the first time " L " appears in the main text to enhance the reader's experience.

Text: " L is the cell length (Classic length 2 mm) (Line 13, Page 7)"

The data in Figure 2b were derived from model analysis, specifically by conducting a mass analysis of the 3D model of the cell unit, resulting in A_{SL1} and A_{LL} . Normalization and dimensionless numbers, A_{SL1}^* and A_{LL}^* , were obtained by dividing

them by L^2 .

$A_{SL1}^* = \frac{A_{SL1}}{L^2}$, and $A_{LL}^* = \frac{A_{LL}}{L^2}$ are dimensionless numbers relating the solid-liquid and the liquid-liquid contact areas, respectively. (Line 16, Page 7)

Q11. Pg7 line 21: “refer to appendix” where in the appendix?

Response: Thank you for the reviewer's inquiry. Our intention was to refer to the Source Data, which is one of the attachments uploaded. The relevant content has been corrected in the manuscript.

Text: (refer to the **Source Data**) (Line 2, Page 8)

Q12. Fig 3:

a. 3b/3d/3f color scheme not consistent n=3 red/orange and n=1 green for both—The original comment was to have the different colors indicate the coordination number across fig 3b, d and f ie n = 1, 2, 3. In fig 3b n =3 is red, while in fig 3d n =1 is blue and in fig 3f n = 3 is green. (I have no problem with purple/other colors; I just think its much easier to follow if the colors indicate the coordination number.)

b. 3d what do the symbols mean? This is not in the main text/figure legend: asterisks denote the working points used throughout the study, circles represent liquid retention, and crosses indicate liquid leakage.

Response: Thank you for the reviewer's suggestions. We have adjusted the color schemes in Figures 3b, d, and f to ensure consistency with the matching scheme of degrees of freedom (n) and colors. Specifically, n=1 corresponds to green, n=2 to

yellow, and $n=3$ to red. Additionally, explanations for the symbols in Figure 3d have been added to the caption (Line 5, Page 11), as shown in the caption below.

Figure 3 (a) Its ability to spontaneously adsorb a liquid and retain it in another incompatible liquid (b) variations in the degrees of freedom of cells at different locations within the reactor (c) the equivalent wettability coefficient λ as a function of D^* (d) the lower limit of the range of values for the equivalent diameter, D_{min}^* , as a function of D^* (asterisks denote the working points used throughout the study, circles represent liquid retention, and crosses indicate liquid leakage) (e) diverse branching configurations of the cellular reactor (f) the distribution of degrees of freedom in the

differently branching cellular reactors

Q13. Table S4: put in more information about the reactors—ie the resin types for reactors C-E

Response: Thank you for the reviewer's suggestion. We have added supplementary information regarding the materials of the cellular reactor to Table S4, aiming to provide additional clarity and detail.

Table S4 Parameters of cellular reactors in this study

Reactor	Structure	Location	L	D*	Blades	N	Resin
Reactor-A		Fig. 3e	2 mm	0.33	3	498	CR
Reactor-B		Fig. 3e	2 mm	0.33	4	568	CR
Reactor-C		Fig. 3e	2 mm	0.33	6	708	CR
Reactor-D			2 mm	0.33	6	708	TR
Reactor-E			2 mm	0.33	6	708	HR
Reactor-F			-	-	6	-	CR
Reactor-G			-	-	-	-	CR
Reactor-H			-	-	-	-	CR

Q14. Table S7: combine with outputs from cellular reactor; hard to compare! Consider combining tables S10/S7, possibly S4? Or putting these results in closer proximity to each other?

Response: Thank you for the reviewer's suggestion. We have updated the sequence of references in the main text and rearranged the figures and tables in the SI accordingly, aligning them with the revised order. Similar-themed figures and tables are now grouped within the same sections, with added section titles to differentiate between the functionalities of each SI part. Additionally, we have compiled the new and old numbering of figures and tables in the SI into a table for improved review efficiency.

Comparison of the new and old numbering of figures in SI.

New	S5	S6	S7	S8	S9	S10	S11	S12
Old	S13	S44	S45	S5	S15	S16	S17	S18
New	S13	S14	S15	S16	S17	S18	S19	S20
Old	S19	S20	S21	S22	S23	S24	S25	S26
New	S21	S22	S23	S24	S25	S26	S27	S28
Old	S27	S28	S29	S30	S31	S32	S33	S34
New	S29	S30	S31	S32	S33	S34	S35	S36
Old	S35	S36	S37	S38	S39	S14	S8	S9
New	S37	S38	S39	S40	S41	S42		
Old	S10	S11	S12	S6	S7	S43		

Comparison of the new and old numbering of tables in SI.

New	S5	S6	S7	S8	S11	S12
Old	S11	S12	S13	S14	S7	S5
New	S13	S14	S15	S16	S17	S18
Old	S6	S18	S8	S15	S16	S17

Q15. Table S12: think the caption should say for entry 21. Seems like inconsistent notation, unless this is describing different reactors? Why is “entry” being used here? Is SA-4 “Ractor-A” in figure S4?

Response: We appreciate the valuable feedback from the reviewer. The original Table S12 (now Table S6) illustrates the catalytic performance of different structural cellular reactors for the same reaction (the addition reaction of dihydrofuran and alcohol). The intention was to continue the entries from items 1-17 as mentioned in the main text. The column containing items 18-21 has now been deleted. "SA-4" refers to "Reactor-A" in Table S4, which has been corrected accordingly, as with the others.

Table S6 Catalytic performance of different fractionated cellular reactors

Reactor	Time 1	Conversion rate 1	Time 2	Conversion rate 2
Reactor -A	60 min	15.4%	120 min	34.6%
Reactor -B	60 min	40.8%	120 min	70.7%
Reactor -C	60 min	50.8%	120 min	83.6%
Reactor -F	60 min	2.48%	90 min	4.86%

REVIEWERS' COMMENTS

Reviewer #2 (Remarks to the Author):

The authors have responded to the concerns I raised. I think the paper is suitable for publication.

Reviewer #3 (Remarks to the Author):

Thank you for addressing the concerns noted. Overall, I am happy with the edits made.

A few minor edits:

main manuscript: pg 12 "eq 9a applies to SC and BCC + frame, while 9b applies to BCC"

SI: pg 74/75 repeated text.

Reviewer #5 (Remarks to the Author):

This was a joint report with reviewer #3.

Reviewer #6 (Remarks to the Author):

This was a joint review conducted with one of the other reviewers; all comments are already contained in the other review.

RESPONSE TO REVIEWERS' COMMENTS

Comments from the Reviewer:

Reviewer: 3

Overall Comments:

Thank you for addressing the concerns noted. Overall, I am happy with the edits made.

A few minor edits:

Q1: main manuscript: pg 12 "eq 9a applies to SC and BCC + frame, while 9b applies to BCC"

Response:

We appreciate the reviewers' attention to detail in the manuscript, and the relevant content has been corrected in the main text.

Equation 9a applies to SC and BCC+frame, while equation 9b applies to BCC. (Line 21, Page 12)

Q2: SI: pg 74/75 repeated text.

Response:

We thank the reviewers for their valuable feedback. There was content overlap between pages 74 and 75 in the Supplementary Information, which has now been addressed by removing the duplication.